# An RNA excited conformational state at atomic resolution

Ainan Geng [1], Laura Ganser[1,6], Rohit Roy [2], Honglue Shi [3,7], Supriya Pratihar[4], David A. Case[5] & Hashim M. Al-Hashimi [4] ✉

Sparse and short-lived excited RNA conformational states are essential players in cell physiology, disease, and therapeutic development, yet determining their 3D structures remains challenging. Combining mutagenesis, NMR spectroscopy, and computational modeling, we determined the 3D structural ensemble formed by a short-lived (lifetime ~2.1 ms) lowly-populated (~0.4%) conformational state in HIV-1 TAR RNA. Through a strand register shift, the excited conformational state completely remodels the 3D structure of the ground state (RMSD from the ground state = 7.2 ± 0.9 Å), forming a surprisingly more ordered conformational ensemble rich in non-canonical mismatches. The structure impedes the formation of the motifs recognized by Tat and the super elongation complex, explaining why this alternative TAR conformation cannot activate HIV-1 transcription. The ability to determine the 3D structures of fleeting RNA states using the presented methodology holds great promise for our understanding of RNA biology, disease mechanisms, and the development of RNA-targeting therapeutics.

With recent breakthroughs in experimental[1] and computational[2] approaches for determining the atomic three-dimensional (3D) structures formed by the most energetically stable ground states (GSs) of biomolecules, the next challenge in structural biology is to determine the 3D structures formed by short-lived and low-abundance conformational states populating local energetic minima along the free-energy landscape[3,4]. These transient, high-energy conformational states, commonly called 'excited conformational states' (ESs)[5], are essential intermediates that form during multistep biochemical reactions, performing functions distinct from those carried out by the more energetically stable GS[3,4]. ESs have also been implicated as drivers of various diseases, and some of them are targets for therapeutic development[3,4,6–10].

Knowing the 3D structures of ESs is essential for deciphering their biological roles and for the rational design of drugs and other biotechnological applications[3,4]. Various methods have been developed to determine the 3D structures of protein ESs, including nuclear magnetic resonance (NMR) spectroscopy[11–14], room-temperature X-ray crystallography[15], and cryo-electron microscopy (CryoEM)[1]. Despite these advancements, including the recent structure determination of a highly populated RNA folding intermediate using CryoEM[16], determining the 3D structures of RNA ESs remains challenging.

Here, we developed a general approach for solving the 3D structural ensemble of RNA ESs, which combines NMR chemical exchange measurements[3,17,18] with targeted mutations stabilizing the minor ES relative to the energetically more stable GS. Our NMR-based approach offers distinct advantages over X-ray crystallography and CryoEM as well as approaches employing ensemble-averaged data[19,20] as it can determine the 3D structures of exceptionally lowly-populated (abundance <1%) and short-lived (lifetime <microsecond) ESs while also measuring their population and lifetime. We developed the approach by determining the conformational ensemble of an ES termed 'ES2'

[1]Department of Biochemistry, Duke University School of Medicine, Durham, NC 27710, USA. [2]Center for Genomic and Computational Biology, Duke University School of Medicine, Durham, NC 27710, USA. [3]Department of Chemistry, Duke University, Durham, NC 27708, USA. [4]Department of Biochemistry and Molecular Biophysics, Columbia University, New York, NY 10032, USA. [5]Department of Chemistry and Chemical Biology, Rutgers University, Piscataway, NJ 08854, USA. [6]Present address: Department of Biophysics, Johns Hopkins University, Baltimore, MD 21218, USA. [7]Present address: Innovative Genomics Institute, University of California, Berkeley, CA 94720, USA. ✉e-mail: ha2639@cumc.columbia.edu

formed by the HIV-1 transactivation response element (TAR) RNA[21–23]. With an exceptionally low population of ~0.4% and a lifetime of ~2.1 ms, the TAR ES2 provides a stringent test for our new methodology.

TAR activates transcription elongation of the HIV-1 retroviral genome by binding to the viral transactivating protein Tat and the super elongation complex (SEC)[24–26]. While no functional role has yet been assigned to the TAR ES2, point-substitution mutations making ES2 the dominant conformation promote kissing-loop dimerization[22], hinting to a potential role in genome dimerization and packaging[27–30] as well as potently inhibit cellular transactivation possibly pointing to a role in releasing Tat-SEC complex[9]. Regardless of its potential functional roles, the 3D structure of the ES2 is of great interest for the design of anti-HIV therapeutics, which inhibit transcriptional activation by preferentially binding and stabilizing this alternative inactive TAR conformation[9,10].

## Strategy for determining conformational ensembles of RNA ESs

Our strategy differs from powerful NMR-based approaches used to determine the 3D structures of protein ESs[11–14], which rely on the chemical exchange to transfer structural information concerning the NMR-invisible ES to the NMR-visible GS, where it can be readily detected. Instead, our approach builds on the observation that RNA ESs typically form by reshuffling base pairs (bps) in and around non-canonical motifs[4,18,21,22,31–34]. Existing NMR methods can determine these alternative secondary structures using chemical shifts, which can be measured even for short-lived, lowly populated states using relaxation dispersion (RD) and chemical exchange saturation transfer (CEST) experiments[18,21–23,31–36] (Fig. 1). The alternative secondary structure then guides the design of a mutant RNA construct, which stabilizes the ES relative to the GS, making it the dominant conformation in solution (Fig. 1). Mutations have also been successfully used to stabilize the ESs formed by proteins[5] and DNA[37], illustrating the versatility of the approach.

The conformational ensemble of the ES-mutant mimic is then determined using Fragment Assembly of RNA with Full-Atom Refinement aided by NMR (FARFAR-NMR)[20,38]. In this recently introduced approach[20], a conformational library is generated for a given NMR-derived RNA secondary structure using FARFAR structure prediction[38] (Fig. 1). The agreement with NMR residual dipolar coupling (RDC)[39,40] data measured for various inter-nuclear bond vectors in the molecule is then used to guide the selection of conformers to be included in an ensemble[39–41] (Fig. 1). RDCs measured between two nuclei report on the orientational distribution of bond vectors relative to a molecule-fixed alignment tensor and are ensemble-averaged over all conformations interconverting on the picosecond to millisecond timescales[42].

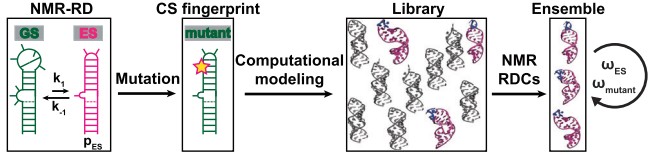

**Fig. 1 | NMR-computational strategy for determining the 3D structural ensemble of excited RNA conformational states.** Chemical exchange experiments are used to elucidate the secondary structure, the exchange kinetics (forward rate $k_1$ and reverse rate $k_{-1}$), and the population ($p_{ES}$) of the ES. Guided by the secondary structure, mutations are used to render the ES the dominant conformation in solution. The ES-mutant is then verified using chemical shift fingerprinting (CS fingerprint). An ensemble for the ES-mutant mimic is then obtained by generating a conformational library for the mutant and using NMR RDCs to select conformers for inclusion into an ensemble. The ensemble is cross-validated against the chemical shifts measured for the ES ($\omega_{ES}$) using chemical exchange experiments as well as for the ES-mutant ($\omega_{mutant}$).

Finally, the RDC-optimized ensemble is cross-validated against ${}^1$H, ${}^{13}$C, and ${}^{15}$N chemical shifts, taking advantage of recent advances in quantum mechanical calculations of chemical shifts given an RNA 3D conformational ensemble[20] (Fig. 1). This key step employs ensemble-averaged chemical shifts measured directly on the ES in the wild-type (wt) RNA molecule. These chemical shift data are exquisitely sensitive to torsion angle and sugar pucker distributions as well as the propensities of bases to stack and hydrogen bond[20,37,43].

## Verifying a mutant mimic of TAR ES2

Utilizing ${}^{13}$C and ${}^{15}$N NMR chemical shifts measured by NMR RD experiments[21], we previously proposed an alternative secondary structure for the TAR ES2, which forms through a strand-register shift that completely remodels the bulge, upper stem, and apical loop, replacing canonical Watson-Crick bps in the GS with a series of closely spaced mismatches (Fig. 2a). Guided by the secondary structure, we previously[22,23] designed a construct (TAR$^{ES2}$), which makes ES2 the dominant conformation by swapping its cUGg$_{syn}$ apical loop with the much more stable cUUCG$_{syn}$g loop (Fig. 2b). This TAR$^{ES2}$ mutant was shown to adopt the alternative ES2 secondary structure as the dominant conformation[22,23]. Moreover, the ${}^1$H, ${}^{13}$C, and ${}^{15}$N chemical shifts measured for the TAR$^{ES2}$ mutant were in quantitative agreement ($R^2 = 0.98$) with those measured for the transient ES2 in wtTAR[21,23] indicating that it is a good structural mimic of this ES (Fig. 2d).

To further confirm that the TAR$^{ES2}$ mutant does indeed mimic the wtTAR ES2 conformational ensemble, we needed to establish that the bps and non-canonical mismatches observed in the TAR$^{ES2}$ mutant also form in the fleeting ES2 (Fig. 2c)[33]. To achieve this, we used the recently introduced high-power SELOPE ${}^1$H CEST experiment[44,45] to measure the guanine and uridine imino ${}^1$H chemical shifts of the transient ES2 in wtTAR, as these chemical shifts are highly sensitive to hydrogen-bonding and base-pairing. We then assessed how well the TAR$^{ES2}$ mutant reproduces these ES2 chemical shifts.

We observed the expected exchange contributions to the ${}^1$H CEST profiles (Fig. 2e and Supplementary Fig. 1a, b) measured for G26, G28, G36, and U38, all of which reshuffle their bp partners when transitioning from the GS into ES2 (Fig. 2a, highlighted in Fig. 2c). Globally fitting the ${}^1$H CEST profiles to a 2-state exchange model yielded a population ($p_{ES2} = 0.25 \pm 0.01\%$) and exchange rate ($k_{ex} = k_1 + k_{-1} = 737 \pm 39$ s$^{-1}$) in very good agreement with values reported previously for ES2 using ${}^{13}$C and ${}^{15}$N RD ($p_{ES2} = 0.40 \pm 0.05\%$ and $k_{ex} = 474 \pm 69$ s$^{-1}$) (Fig. 2f). The imino ${}^1$H chemical shifts determined for ES2 using ${}^1$H CEST were in excellent agreement (RMSD = 0.2 ppm) with counterparts measured for the TAR$^{ES2}$ mutant (Fig. 2g). These results reinforce the validity of TAR$^{ES2}$ as an ES2-mimic and substantiate formation of Watson-Crick G-C, wobble U-U, and two Watson-Crick G$_{anti}$-A$_{anti}$ mismatches in the transient ES2 (Fig. 2a, c), greatly facilitating 3D structure determination. They also establish the utility of high-power ${}^1$H CEST experiment in studying RNA ESs.

## Measurement of residual dipolar couplings

Having verified that the TAR$^{ES2}$ mutant mimics ES2, we determined its conformational ensemble using FARFAR-NMR[20,38]. We measured one-bond ${}^{13}$C-${}^1$H ($^1D_{CH}$) and ${}^{15}$N-${}^1$H ($^1D_{NH}$) RDCs in Pf1 phage (~17 mg/ml)[46] in TAR$^{ES2}$ as well as on an elongated variant (E-TAR$^{ES2}$) in which the lower helix was extended by five bps (Fig. 3a, Supplementary Fig. 2, 3a, Supplementary Table 1). The elongation was used to modulate alignment and to obtain an additional RDC dataset for ensemble determination[47,48]. Two independent frequency-based experiments were used to obtain splittings encoded along the ${}^{13}$C/${}^{15}$N or ${}^1$H dimensions[47], respectively. The root-mean-square-deviation (RMSD) between the two sets of measurements (~2.0 Hz) was used to estimate the RDC uncertainty (Supplementary Fig. 3b).

The RDCs measured for TAR$^{ES2}$ differed markedly (RMSD = 14.3 Hz) from counterparts measured in wtTAR, indicating that the ES2 and GS

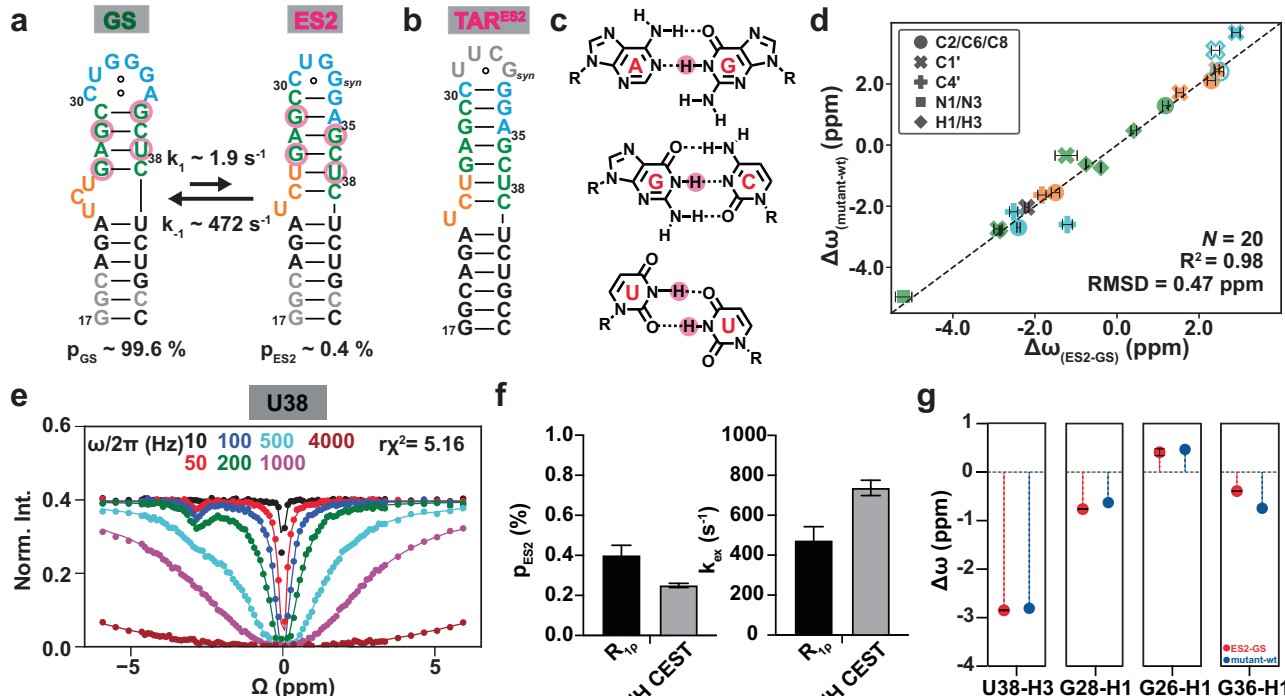

**Fig. 2 | Verifying a mutant mimic of the HIV-1 TAR ES2 using chemical shift fingerprinting. a** The TAR GS exists in dynamic equilibrium with a sparsely populated and short-lived excited conformational state, ES2. The secondary structure of ES2 was deduced based on prior [13]C and [15]N chemical shifts[21,23] obtained using NMR $R_{1\rho}$ experiments for residues throughout the lower helix, bulge, upper stem, and apical loop (shown in color). Residues showing exchange contributions in the [1]H CEST experiment are circled in pink. **b** The TAR[ES2] mutant[10,22,23] stabilizes an ES2-like conformation as the dominant GS. **c** The base pairs and mismatches in ES2 which were verified using the [1]H CEST experiments. **d** Correlation plot comparing the difference between the [1]H, [13]C, and [15]N chemical shifts measured between the TAR ES2 and GS ($\Delta\omega_{(ES2-GS)} = \omega_{ES2} - \omega_{GS}$) using NMR $R_{1\rho}$ ([13]C and [15]N) and CEST ([1]H) with the corresponding difference in chemical shifts ($\Delta\omega_{(mutant-wt)} = \omega_{mutant} - \omega_{wt}$) obtained from comparing the GS chemical shifts for TAR[ES2] and wtTAR. For resonances belonging to G33 near the site of mutation (in open symbols), values were derived from the G28U TAR ES2 mutant[21]. $\Delta\omega$ are color-coded according to the structural elements in (a). The $N$ represents the total number of NMR probes used for the comparison in the correlation plots. **e** Representative [1]H CEST profile for U38-H3 showing an exchange contribution. RF field powers used are color-coded. Remaining data are shown in Supplementary Fig. 1a. **f** Comparison of the population ($p_{ES2}$) and kinetic exchange rate ($k_{ex} = k_1 + k_{-1}$) obtained by [1]H CEST with values measured previously[21] using [13]C, [15]N $R_{1\rho}$. $R_{1\rho}$ or CEST measurement data presented here are the mean values ± 1 s.d. from Monte Carlo simulations (number of iterations = 500) as described in Methods. The errors in exchange parameters derived from [1]H CEST were set equal to the fitting errors determined as the square root of the diagonal elements of the covariance matrix. **g** Comparing the difference between the [1]H chemical shifts measured between the TAR ES2 and GS ($\Delta\omega = \omega_{ES2} - \omega_{GS}$) using [1]H CEST (in red) with the corresponding difference in chemical shifts ($\Delta\omega = \omega_{mutant} - \omega_{wt}$) obtained from comparing the GS chemical shifts for the mutant TAR[ES2] and wtTAR (in blue). Error bars denote the error of exchange parameters.

form different conformations (Supplementary Fig. 3c). The similar RDCs measured in TAR[ES2] and E-TAR[ES2] (Supplementary Fig. 3d) and for the two TAR[ES2] helices (Supplementary Fig. 3e) indicated that they are not substantially kinked relative to one another or undergoing large amplitude inter-helical motions across the single uridine bulge. This was in stark contrast to the TAR GS (Supplementary Fig. 3f), in which collective inter-helical motions about the trinucleotide bulge resulted in markedly different RDCs upon helix-elongation as well as differential attenuation of the RDCs measured in the two helices[47,49,50]. Thus, the remodeling of junction topology and shortening of the bulge linker appears to alter the TAR global conformation likely reducing the amplitude of inter-helical motions. Nevertheless, the attenuated RDCs and downfield shifted aromatic U23-C6 chemical shift (Supplementary Fig. 2) indicated that the bulge residue U23 remains locally flexible in TAR[ES2].

### Determining the ES2 conformational ensemble using FARFAR and RDCs

We used FARFAR-NMR to determine ensembles of the ES-mutant by integrating FARFAR structure prediction with NMR RDC data and then used chemical shifts to cross-validate the generated ensemble. Using

FARFAR, we generated a conformational library of $N = 10,000$ conformers given the NMR-derived TAR[ES2] secondary structure (Fig. 3a). Ensemble averaging over the entire library resulted in poor agreement with the two sets of RDCs; the RMSD of 10.2 Hz substantially exceeded the experimental uncertainty of 2.0 Hz (Fig. 3b and Supplementary Fig. 4a). No single conformer in the FARFAR library satisfied the RDCs and poor agreement was also obtained when ensemble averaging over the ten lowest energy conformations based on the Rosetta energy score (RMSD = 11.2 Hz and $R^2 = 0.71$, Fig. 3b and Supplementary Fig. 4b, e) or ten conformations selected randomly (RMSD = 10.7 Hz and $R^2 = 0.74$, Fig. 3b and Supplementary Fig. 4c).

Using sample and select (SAS)[41], we used the agreement with the two sets of measured TAR[ES2] RDCs to guide the selection of a subset of conformers from the FARFAR library to form an optimized FARFAR-NMR ensemble. Testing increasingly larger ensemble sizes ($N$), starting with $N = 1$ up to $N = 49$, an optimal ensemble with $N = 10$ conformers (see Supplementary Fig. 4f) could be obtained, which showed improved RDC agreement across both helices and the bulge. However, despite RDC optimization, the RMSD = 3.3 Hz still exceeded experimental uncertainty (Fig. 3b and Supplementary Fig. 4d). Moreover, cross validation of the FARFAR-NMR ensemble by using the AF-QM/

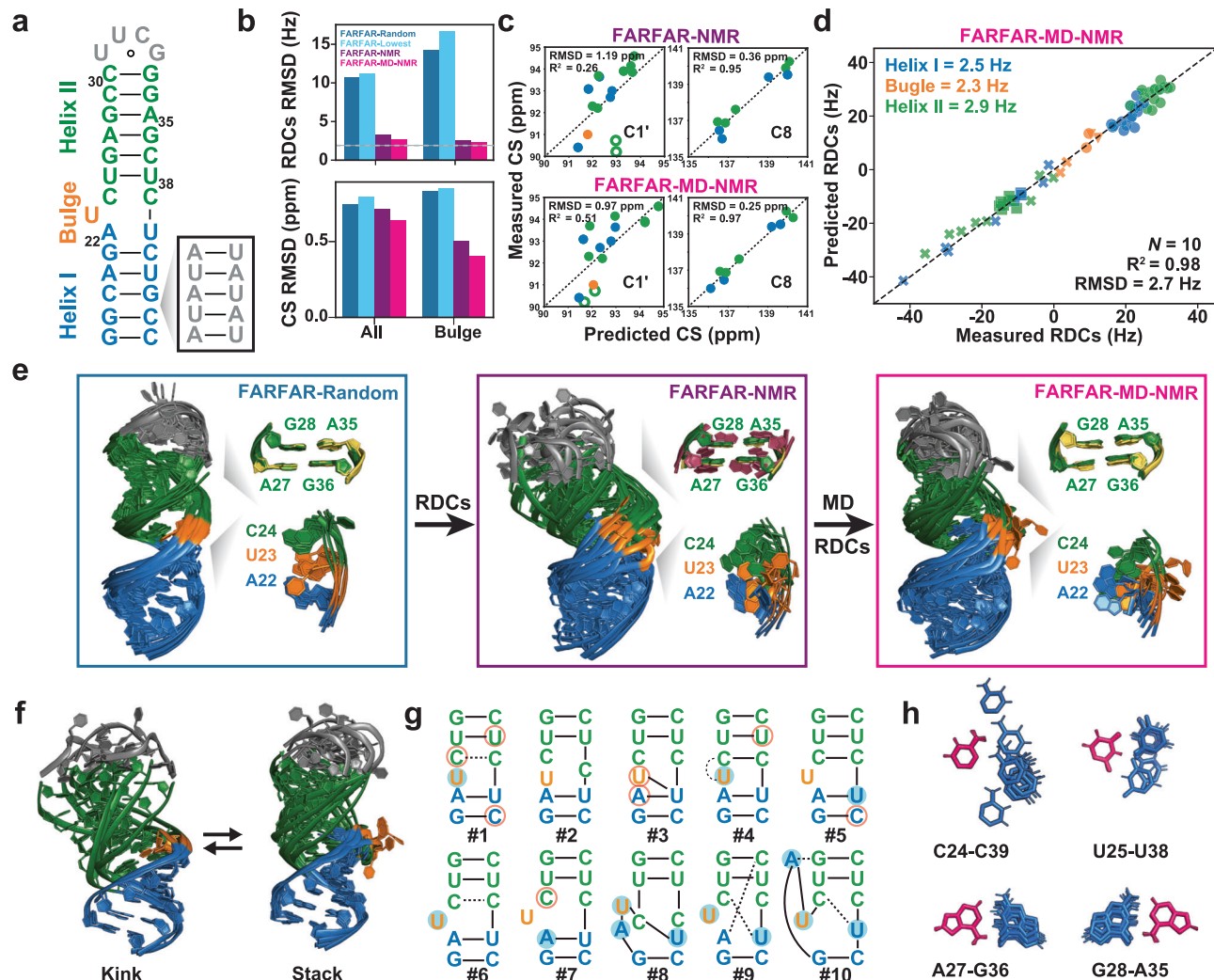

**Fig. 3 | Determining the conformational ensemble of TAR^ES2.** **a** Secondary structure of TAR^ES2 and an elongated construct (E-TAR^ES2) obtained by inserting residues (in gray) at the indicated position in the lower helix. **b** Comparison of the RMSD between measured and predicted RDCs (top, TAR^ES2 + E-TAR^ES2, dashed line corresponds to uncertainty) and chemical shifts (bottom, measured on wtTAR ES2) for the FARFAR-Random (deep blue), FARFAR-Lowest (light blue), FARFAR-NMR (purple), and FARFAR-MD-NMR (magenta) ensembles. Bulge refers to chemical shifts measured at residue U23. **c**, **d** Comparison between measured and ensemble predicted (**c**) representative chemical shifts measured on wtTAR ES2 (The outlier A-C1′ chemical shifts are highlighted in open green symbols) and (**d**) RDCs (TAR^ES2 + E-TAR^ES2). Also shown is the number of conformers (N) in the ensemble as well as the R^2 and RMSD between measured and predicted values. RDCs and chemical shifts are color-coded according to the structural elements in (**a**). **e** Structural overlay of

the FARFAR-Random, FARFAR-NMR, and FARFAR-MD-NMR ensembles (N = 10) along with a zoomed-in view of the tandem AG mismatches and bulge motif. Partially melted, high-χ angle (−130 ± 20°), and regular χ-angle (<−150°) A-G mismatches are colored purple, yellow, and green, respectively. **f** Overlay of conformers showing inter-helical stacking accompanied by the flipping out of bulge residue U23. **g** Secondary structure of the ten conformers in the FARFAR-MD-NMR ensemble. Residues with C2′-endo sugar pucker and non-gauche+ γ (falling outside 20-100°) torsion angle are highlighted using blue filled and orange open circles, respectively. Watson–Crick bp and Wobble bp are denoted using a solid line, whereas other bp geometries are indicated as a dashed line. **h** Structural overlay of mismatches in the FARFAR-MD-NMR ensemble with bases used as a reference for superposition colored pink.

MM[20,43] approach to predict ensemble-averaged chemical shifts (Fig. 3b and Supplementary Figs. 5a–f, 6a–f) revealed that the ensemble poorly predicted the upfield shifted A27-C1′ and A35-C1′ chemical shifts (Fig. 3c), which form tandem Watson-Crick G_anti-A_anti mismatches in TAR^ES2 (Fig. 3a).

**Optimizing ensemble using MD**
Inspection of the FARFAR ES2 library revealed that it was dominated by conformations in which bulge residue U23 and its neighboring residues (A22, C24, and U25) are intra-helical and stacked, with their sugar moieties primarily adopting the canonical C3′-endo sugar pucker (Supplementary Fig. 7a, b). And yet the attenuated RDCs and downfield shifted U23-C6 chemical shift indicated that the bulge residue is flexible, and a prior analysis of ^3J_{H1′H2′} scalar couplings and C1′ and C4′

chemical shifts indicated that A22 and U23 significantly sample the non-canonical C2′-endo sugar pucker[23].

FARFAR relies on fragments from the crystallographic database to build RNA structural models[38]. The unique closely spaced noncanonical motifs found in ES2 may be poorly represented in the PDB and thus difficult to model using this fragment-based approach. Therefore, to increase the conformational diversity and refine the ensemble further, we subjected the ten TAR^ES2 conformers in the optimized FARFAR-NMR ensemble to 600 ns MD simulations using the RNA OL3 force field[51]. SAS optimization of the MD-generated conformational library yielded an N = 10 ensemble (FARFAR-MD-NMR) (Supplementary Movie 1), which robustly showed improved agreement with both the RDCs (RMSD = 2.7 Hz) (Fig. 3b, d) and chemical shifts (Fig. 3b, c and Supplementary Figs. 5d, 6d).

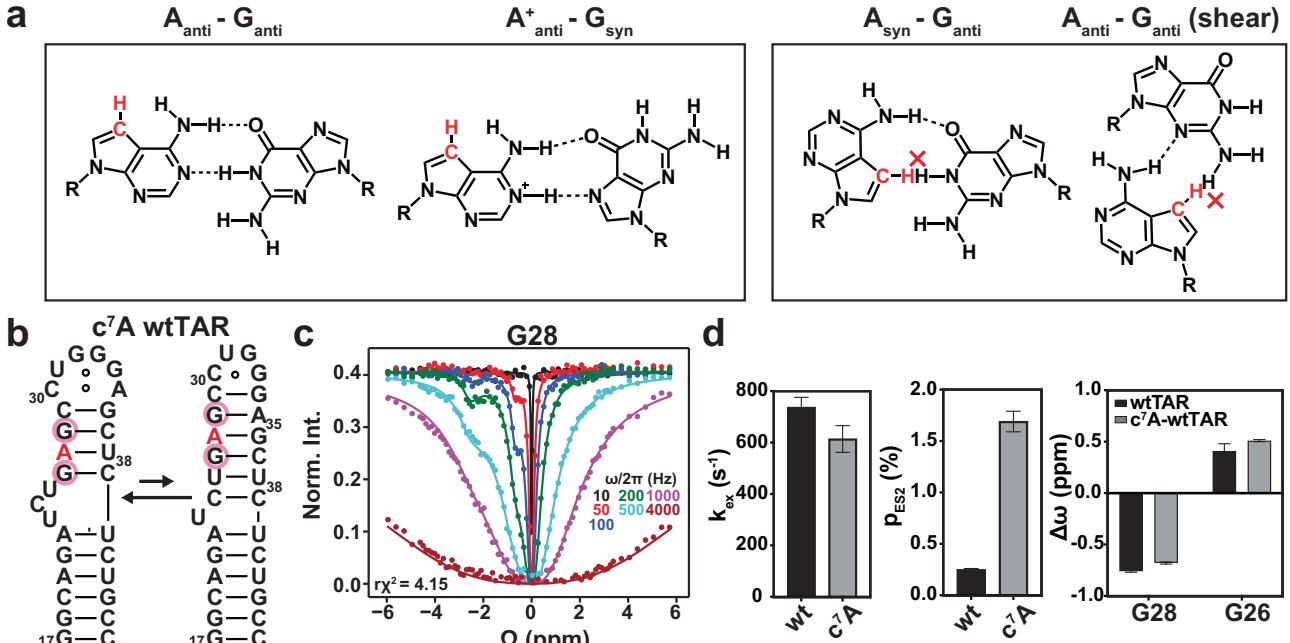

**Fig. 4 | Cross-validating the A27$_{anti}$-G36$_{anti}$ conformation in ES2 using single-atom substitutions. a** The c$^7$A substitution of A27 is predicted to selectively destabilize A$_{syn}$-G$_{anti}$ Hoogsteen and sheared A-G conformations but not the A$_{anti}$-G$_{anti}$ conformation formed in the TAR$^{ES2}$ ensemble. **b** c$^7$A wtTAR preserves the dynamic equilibrium between the GS and ES2. The modified residues are highlighted in red, and residues showing exchange contributions to ES2 in the $^1$H CEST experiment are circled in pink. **c** Representative $^1$H CEST profile for G28-H1 showing an exchange contribution. RF field powers used are color-coded. Remaining data are shown in Supplementary Fig. 10. **d** Comparison of the kinetic exchange rate (k$_{ex}$ = k$_1$ + k$_{-1}$), the population (p$_{ES2}$) and chemical shift difference (Δω = ω$_{ES2}$ − ω$_{GS}$) measured on wtTAR and c$^7$A wtTAR. Data presented here are the mean values ± standard error (SEM) derived from the square root of the diagonal elements in the covariance matrix of the fitted parameters.

The optimized FARFAR-MD-NMR ensemble included conformations in which U23 was flipped out and in which the two helices were coaxially stacked (Fig. 3e). Coaxial stacking of helices coupled to the flipping out of intervening bulge residues (Fig. 3f) is commonly observed in RNAs including in the TAR GS[20]. In contrast, not only were conformers with U23 flipped out rare in the FARFAR library, but those selected in the RDC optimized FARFAR-NMR ensemble had neighboring bps that were partially melted, and the helices were not coaxially stacked (Fig. 3e). Excluding conformations with U23 flipped out from the FARFAR-MD library reduced the RDC agreement (RMSD = 4.4 Hz) to a level comparable to that of FARFAR-NMR (RMSD = 3.3 Hz) (Supplementary Fig. 7c). Thus, the coaxial conformations with U23 flipped out likely accounted for the improved RDC agreement obtained with the FARFAR-MD versus FARFAR library.

The FARFAR-MD-NMR ensemble also better modeled the tandem G-A mismatches relative to FARFAR-NMR, leading to improved predictions of the A27-C1' and A35-C1' chemical shifts (Fig. 3d, Supplementary Figs. 5c–f, 6c–f). A27 and A35 robustly formed tandem Watson-Crick G$_{anti}$-A$_{anti}$ mismatches in the FARFAR-MD-NMR ensemble, whereas some of these mismatches were partially melted in the FARFAR-NMR ensemble (Fig. 3e). In addition, the A27 glycosidic χ-angle was high (−142-146°) for several conformers (Fig. 3e and Supplementary Fig. 7d), in agreement with a prior analysis of the upfield shifted C1' (but not C4') ES2 chemical shifts[23], and these conformers were associated with upfield shifted A27-C1' and A35-C1', resulting in better agreement with the experimentally measured values (Fig. 3e). Omitting these conformers from the FARFAR-MD-NMR ensemble resulted in poorer agreement with the chemical shifts (Supplementary Fig. 7e, f). High adenosine χ-angles were also previously reported in solution NMR structures of tandem AG mismatches[52]. In contrast, in the FARFAR-NMR ensemble, none of the A27 χ-angles were as high as ~−140° (Fig. 3e).

The FARFAR-MD-NMR ensemble also included alternative secondary structures in which U23 or C24 pair with U40 (Fig. 3g). Due to their propensity to flip out, A22 and U23 were enriched in non-canonical C2'-endo sugar pucker (Supplementary Fig. 7b), in excellent agreement with the sugar pucker distributions, deduced independently using $^3$J$_{H1'H2'}$ scalar couplings and C1' and C4' chemical shifts[23]. In both the FARFAR-NMR and FARFAR-MD-NMR ensembles, U25-U38 formed alternative wobble conformations, while C24-C39 sampled a wide range of partially paired and unpaired conformations (Fig. 3h) in good agreement with the downfield shifted C24-C6 and C39-C6 chemical shifts and interrupted H8/6-H1' NOE connectivity at C24-C39 and U25-U38 (Supplementary Figs. 2, 8, 9).

### Cross-validating the ensemble using single-atom substitutions

Based on our FARFAR-NMR and FARFAR-MD-NMR ensembles, the tandem AG mismatches adopt the A$_{anti}$-G$_{anti}$ conformation, not other commonly observed conformations such as the A$_{syn}$-G$_{anti}$ Hoogsteen and sheared A-G (Fig. 4a). Thus, our ensemble predicts that replacing the adenosine base with its isosteric base analog 7-deaza-adenosine (c$^7$A) that replaces N7 by C7H7 should not impact the formation of the ES2 (Fig. 4a); on the other hand, based on our prior work on DNA[53], the modification should destabilize and potentially quench exchange with ES2 if the AG mismatch in ES2 adopted either the A$_{syn}$-G$_{anti}$ Hoogsteen or sheared A-G conformation (Fig. 4a). We tested this prediction using chemical synthesis to prepare wtTAR in which A27 is substituted with c$^7$A (Fig. 4b). The G28-H1 and G26-H1 $^1$H CEST profiles could be combined in a global fit, yielding downfield shifted G28-H1 (Δω of ~−0.7 ppm) and upfield shifted G26-H1 (Δω of -0.5 ppm) chemical shifts consistent with the Watson-Crick A$_{anti}$-G$_{anti}$ bps in the wtTAR ES2 (Fig. 4c, d and Supplementary Fig. 10a, c). The exchange rate of k$_{ex}$ = 614 ± 52 s$^{-1}$ was in very good agreement with values measured for the wtTAR ES2 (k$_{ex}$ = 737 ± 39 s$^{-1}$) using $^1$H CEST

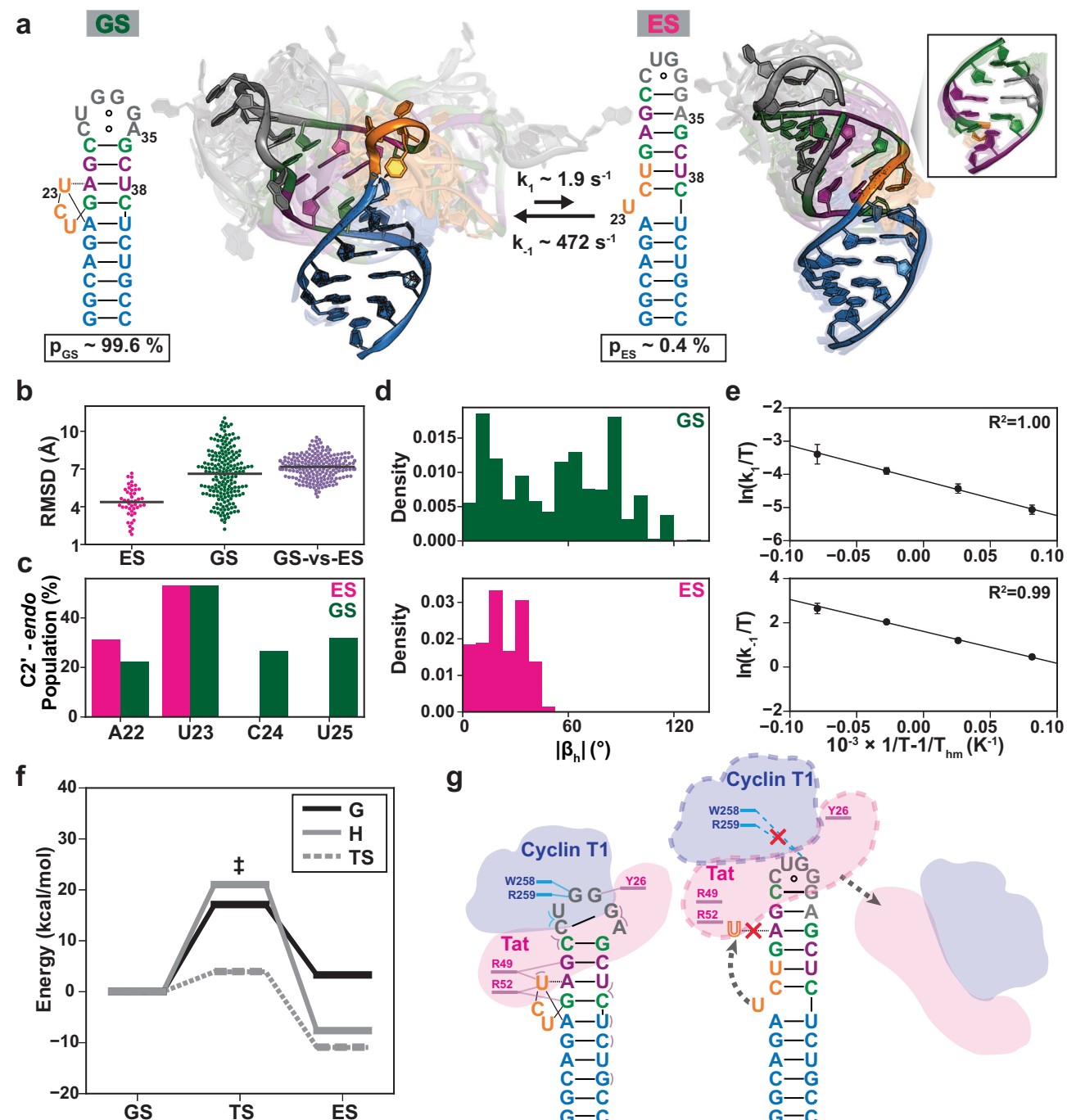

**Fig. 5 | Comparison of the 3D structure and energetics of the TAR GS and ES2.**
**a** Comparison of the secondary structure and 3D conformational ensemble of the TAR GS and ES2 along with the kinetic rates of interconversion and equilibrium populations. **b**–**d** Comparison of the breadth of the GS[20] and ES2 ensemble distributions using **b** heavy-atom pairwise RMSD (Methods) **c** sugar pucker distributions and **d** inter-helical bend angles ($\beta_h$). $|\beta_h|$ is the absolute magnitude of the bend angle. The horizonal black line in (**b**) represents the average RMSD within each distribution. **e** Temperature-dependent forward ($k_1$) and reverse ($k_{-1}$) rate constants for the GS to ES2 exchange obtained using $^1$H CEST experiments performed on

residue U38-H3 in wtTAR. Error bars in $k_1$ and $k_{-1}$ were determined by propagating the errors in exchange parameters obtained from 2-state fits of the $^1$H CEST profile for U38-H3 to the Bloch-McConnell equations (Methods). **f** The kinetic-thermodynamic profile for exchange in wtTAR between the GS and ES2 via a transition state (‡), showing activation and net free energy (G), enthalpy (H), and entropy (TS) changes with GS referenced to 0. **g** Schematic illustration of how TAR$^{ES2}$ interrupts specific interactions with both Tat (pink) and Cyclin-T1 (blue) by disrupting the formation of base-triple involving U23 and remodeling the apical loop.

(Fig. 4d). Indeed, the modification did not quench the exchange rather it increased the population of the ES2 by ~8-fold, possibly due to the destabilization of the GS[53]. Similar results were obtained robustly for the HIV-2 TAR variant (2U) with the UU dinucleotide bugle (Supplementary Fig. 10b, d).

## ES2 differs substantially from the GS and is more ordered

The TAR ES2 ensemble offered a unique glimpse into the 3D atomic structure formed by a high-energy RNA conformational sub-state (Fig. 5a). This structure diverges significantly from the GS (Fig. 5a), with an average heavy-atom RMSD of 7.2 ± 0.9 Å (Fig. 5b). Not only do the

GS and ES2 have distinct secondary structures (Fig. 5a), but they also vary considerably in their global shapes (Fig. 5a). In the GS, the helices adopt a wide range of inter-helical orientations spanning stacked and kinked conformations (Fig. 5a). Inter-helical stacking is accompanied by extra-helical flipping and changes in the sugar pucker distribution at the three bulge residues (Fig. 5b, c). Conversely, in ES2, the two helices consistently maintain co-axial stacking, sampling a much narrower distribution of inter-helical orientations (Fig. 5d), with U23 flipping in and out in a manner coupled to changes in the sugar pucker distribution (Fig. 5c). Thus, the two ensembles exhibit differences one might expect when comparing unrelated RNA sequences.

It would be reasonable to expect that a high-energy conformational state such as ES2 would lose native contacts and form a broader ensemble distribution relative to the energetically more favored GS. Yet based on heavy-atom RMSD (Fig. 5b), sugar pucker distributions (Fig. 5c), and global inter-helical orientation (Fig. 5d), the ES2 ensemble was much more ordered than the TAR GS (Fig. 5a). Despite being rich in non-canonical mismatches, the upper stem in TAR[ES2] forms a highly structured helix in which the mismatches are base-paired in most conformations. When excluding the C24-C29 terminal bp, the upper stem superimposes with an idealized A-form helix with heavy-atom RMSD of $1.4 \pm 0.2$ Å (Fig. 5a). On aggregate, TAR[ES2] also has a larger number of bps relative to the GS.

**ES2 is entropically disfavored relative to the GS**
Our findings raised the question: why is the highly structured ES2 less energetically favorable than the GS? Indeed, many secondary structure prediction programs predict ES2 to be the more stable conformation. To further dissect the relative thermodynamic stabilities of the GS and ES2, we used $^1$H CEST experiments to measure the temperature-dependence of the GS to ES2 exchange in wtTAR. Analysis of the temperature-dependent exchange parameters (Fig. 5e, f and Supplementary Fig. 11a) revealed that ES2 is enthalpically favored relative to the GS by $\Delta H_{ES2-GS} = -7.7 \pm 1.3$ kcal/mol (Fig. 5f); a result which we confirmed using $^{15}$N CEST and off-resonance $R_{1\rho}$ experiments (Supplementary Figs. 11b, c, 13a, b). In addition, analogous experiments on the TAR[ES2] and G36U mutants (Supplementary Figs. 12–14), both of which back exchange with a GS-like conformation[10], yielded oppositely signed $\Delta H_{GS-ES2}$. Thus, ES2 is less energetically favorable relative to the GS due to the loss of favorable entropy. Although the origins of this entropy difference (e.g., solvent, metal ions, conformation, etc.) remain to be dissected, the loss of conformational entropy when transitioning into the more structured ES2 ensemble could be an important contribution.

## Discussion

Previous studies have demonstrated the utility of mutations to stabilize sparsely populated ESs in functionally diverse RNAs[18]. These ESs include folding intermediates[32,33], conformations recognized by the microRNA processing machinery[34], and inactive conformations, considered attractive drug targets[31,32]. The validity of these mutants as ES-mimics could be further verified in the future through extensive measurements of relaxation dispersion data across various nuclei and residues, and this in turn, could provide a route for solving the 3D structures of these other RNA ESs. These applications are needed to test the general applicability of the approach on diverse RNAs. Interestingly, based on secondary structure, many of the ESs formed by other RNAs are also predicted to be more ordered than the GS[31,32,36]. Thus, FARFAR-NMR could reveal more intricate and structured conformational states populating higher-energy levels of the RNA-free energy landscape.

Although the TAR[ES2] mutant recapitulated the behavior of the bulge and upper stem in the ES2, it utilized a non-native apical loop to stabilize this high-energy conformation. Consequently, our ensemble did not provide insights into the ES2 apical loop, shown previously to promote TAR dimerization[22]. The FARFAR-NMR approach could be used in the future to determine the structure of another ES2 mutant, which replaces the G-A mismatch with a U-A bp[9,21] thus preserving the native ES2 apical loop. Alternatively, RDCs[54] and other structural constraints[14] could be measured directly for the transient ES2 using chemical exchange-based methods[54] and used in FARFAR-NMR ensemble determination.

The TAR ES2 ensemble highlights the remarkable conformational diversity of motifs rich in non-canonical mismatches commonly observed in RNA ESs[18]. Based on a prior structure survey[55], A-G mismatches flanked by Watson-Crick bps predominantly form the Watson-Crick $A_{anti}$-$G_{anti}$ conformation. Prior NMR structures[52] have also demonstrated that tandem A-G mismatches within the AG sequence context, flanked by Watson-Crick bps, also form the $A_{anti}$-$G_{anti}$ conformation, with adenosines having high glycosidic angles[52], but that they can also form $A_{anti}$-$G_{anti}$ or the sheared conformation in the GA sequence context depending the identity of the flanking Watson-Crick bps[56,57]. The sheared conformation is also observed robustly in X-ray structures of tandem A-G mismatches in different sequence contexts when they are near non-canonical motifs or near sites forming intermolecular contacts[58] (Supplementary Table 2). The lack of X-ray structures for tandem A-G mismatches in the AG sequence context, sandwiched by Watson-Crick bps, as occurs in ES2, might explain the canonical glycosidic angles for the adenosines in the FARFAR-generated models (Fig. 3e). In addition to sequence context, the preferences to form alternative A-G conformations can also be modulated by post-transcriptional modifications[59]. Thus, additional studies are needed to dissect the contextual and environmental effects modulating the ensemble behavior of motifs rich in non-canonical mismatches, which appear to be prevalent in RNA ESs.

Our approach for determining conformational ensembles of the ES-stabilizing mutant relies on using state-of-the-art modeling tools including FARFAR and MD simulations to generate an initial library of conformers then using the RDCs to select conformations in the ensemble and chemical shifts to test the ensembles. Compared to conventional structure determination protocols, this approach treats ensemble averaging of the NMR data, uses physical models to address the inherent degeneracies when solving ensembles; and has the advantage of testing state-of-the-art models of RNA structure guiding their future development. In particular, in our prior work on the TAR GS[20], we showed that FARFAR does a superior job sampling sugar pucker conformations relative to MD simulations whereas the current work highlights the advantages of using MD in modeling tandem G-A mismatches which may underrepresented in the PDB. However, because RNA ESs such as ES2 are likely to have unusual motifs that are underrepresented in the PDB, it may be helpful and, in some cases, even necessary to pursue full-fledged structure determination and to use the resulting structures as starting points for generating ensembles. Alternatively, multi-conformer refinement approaches could also be used to determine ensembles for the ES-stabilizing mutants[60,61].

Our findings also have important implications for RNA structure prediction and efforts to rationally control and engineer RNA behavior. For TAR, the ES2 was enthalpically favored over the GS, was more structured, and had a larger number of bps and mismatches. The greater entropic stability of GS does not appear to originate from interactions with metal ions, as prior studies showed that adding 1 mM $Mg^{2+}$ minimally affects the TAR GS-ES2 exchange kinetics and thermodynamics[62]. Because GS forms a broader conformational ensemble relative to the ES2, the greater stability of the GS may be driven by conformational entropy. This suggests that a conformational ensemble description may ultimately be required to accurately predict RNA 3D structure and to discriminate the GS from competing ESs.

The non-native highly structured ES2 ensemble explains why it does not support HIV-1 transcriptional activation[9]. Not only is ES2

incapable of forming the base-triple motif required for Tat binding[25], but the shape of the apical loop, which contacts both Tat and Cyclin-T1[26], is also substantially altered (Fig. 5g) relative to the GS. Moreover, the structure increases the spacing between the bulge and the apical loop, potentially disrupting the simultaneous engagement of Tat and Cylcin-T1 (Fig. 5g). These attributes make ES2 an attractive target for developing anti-HIV therapeutics, which inhibit transcriptional activation.

Thus far, efforts targeting the TAR GS with small molecules have failed to yield potent and selective inhibitors of HIV-1 transcriptional activation[63]. Not only is it challenging to find compounds that can compete with the Tat-SEC complex for TAR binding, achieving the desired binding selectivity is also difficult because the GS is predominantly composed of canonical Watson-Crick bps, which are abundant in the transcriptome[63]. On the other hand, Tat-SEC cannot productively bind ES2[9], and fewer sequences are likely to adopt 3D structures like ES2. Therefore, it may be possible to enhance the selectivity and potency of small molecule inhibitors of TAR by optimizing them to preferentially bind ES2 over the GS. A recent proof-of-concept study demonstrated that a ligand could selectively bind to a sparsely populated (-13%) RNA conformational state and make it the dominant conformation[10]. One strategy would involve subjecting the 3D structural ensemble of TAR ES2 and GS[20] to virtual screening[64] and identifying compounds that are predicted to preferentially bind ES2. This approach could be extended to other RNA drug targets that adopt inactive ES conformations[31,32]. Thus, the methodology presented here holds great promise in illuminating the functional roles of RNA ESs and advancing methods to exploit them in biotechnological applications.

While no functional role has been assigned to the TAR ES2, we speculate that ES2 could play a role in the dimerization and packaging of the retroviral genome. It has been shown that TAR is required for proper genome dimerization and/or packaging through mechanisms that are not fully understood[27–29]. In addition, we previously showed that the TAR ES2 has a high propensity to form kissing dimers[9,22]. Finally, deletion of the UCU trinucleotide bulge, which inhibits the formation of ES2[21], significantly impairs retroviral genome dimerization[30]. These functional roles can be tested in the future by examining the consequence of introducing TAR ES2-stabilizing and ES2-destabilizing mutations in dimerization and packaging assays.

## Methods
### RNA preparation
Unlabeled wtTAR, TAR$^{ES2}$, G36U mutant, c$^7$A wtTAR, and c$^7$A 2U RNA were synthesized using a MerMade 6 Oligo Synthesizer (BioAutomation) using standard phosphoramidite chemistry and base and 2′-hydroxyl deprotection protocols as described previously[62]. Unlabeled phosphoramidites were purchased from ChemGenes. Uniformly $^{15}$N/$^{13}$C labeled wtTAR, TAR$^{ES2}$, and E-TAR$^{ES2}$ were prepared by in vitro transcription using T7 RNA polymerase (New England BioLabs), synthetic DNA template (Integrated DNA technologies) containing the T7 promoter sequence (TTAATACGACTCACTATA), and uniformly labeled $^{15}$N/$^{13}$C nucleotides (Cambridge Isotope Laboratories, Inc.). The transcription reaction was carried out at 37 °C for 16 h. All RNAs were purified using a 20% (w/v) denaturing polyacrylamide gel with 8 M urea and 1X TBE (Tris/borate/EDTA). The RNA was extracted from the excised gel by electro-elution (Bio-Rad) followed by concentration and ethanol precipitation. The RNA was then annealed in water at 95 °C for 5 min and snap-cooled on ice for 1 h. Finally, RNA was buffer exchanged using an Amicron Ultra-15 centrifugal filter into NMR buffer (15 mM sodium phosphate, 25 mM sodium chloride, 0.1 mM EDTA and pH 6.4). 10% (v/v) D$_2$O was added to each sample before NMR data collection. The final concentration of RNA samples ranged between 0.8 and 1.4 mM.

## NMR experiments
NMR experiments were carried out on Bruker Avance III 600-MHz, Bruker Avance TS2.1 800 MHz, Bruker Avance TS2.1 900 MHz, and NEO 900 MHz spectrometers equipped with 5-mm triple-resonance cryogenic probes. NMR data was analyzed using NMRPipe[65] and SPARKY (T.D. Goddard and D.G. Kneller, SPARKY 3, University of California, San Francisco). All experiments were performed in NMR buffer with 15 mM sodium phosphate, 25 mM NaCl, 0.1 mM EDTA at pH 6.4 and 10% D$_2$O, unless stated otherwise.

**Resonance assignment.** NMR chemical shift assignments for exchangeable and nonexchangeable protons in TAR$^{ES2}$ were obtained from prior studies[22,23]. To expand and verify these prior resonance assignments, we measured the 2D HCN experiment on labeled TAR$^{ES2}$ at 15 °C in 100% D$_2$O on a 600 MHz Bruker spectrometer equipped with HCN cryogenic probes. This led us to update a few resonance assignments relative to the prior study[22] (Supplementary Fig. 9). The resonance originally[22] assigned as U25-C6H6 was updated to U38-C6H6. This update did not impact the prior study because no RD measurements were performed on this probe. The resonance originally[22] assigned as C39-C6H6 was updated to U25-C6H6 leading to a stronger correlation between the Δω value for U25-C6 deduced from the wt and mutant chemical shifts and the value measured using relaxation dispersion[22]. Finally, the updated C39-C6H6 resonance is now overlapped with G54-C8H8, which again does not impact our prior study because Δω was never measured for this probe using $R_{1\rho}$. The new resonance assignments have been deposited to the BMRB database[66]. In addition, we were able to assign the imino resonances of G28 and G36 in the tandem AG mismatch motif by collecting low temperature (5 °C) 2D $^1$H–$^1$H nuclear Overhauser effect spectroscopy (NOESY) experiment with mixing time of 200 ms (Supplementary Fig. 8a, b).

**Measurement of RDCs.** One-bond C-H ($^1$D$_{CH}$) and N-H ($^1$D$_{NH}$) RDCs were measured at 25 °C using a 600 MHz Bruker spectrometer equipped with a 5-mm triple-resonance cryogenic probe. C2H2, C6H6, C8H8, C5H5, and C1′H1′ splittings were measured along $^1$H dimension using 2D transverse relaxation-optimized spectroscopy (TROSY) experiment and along the $^{13}$C dimension using the 2D $^1$H-$^{13}$C S$^3$CT-heteronuclear single-quantum correlation (HSQC) experiment[67]. N-H (N1H1, N3H3) splittings were measured using 2D $^1$H/$^{15}$N HSQC experiments in duplicate without decoupling in the indirect ($^{15}$N) or direct dimensions ($^1$H)[47]. RDCs were measured as the difference between splittings obtained in the absence (J) and presence (J + D) of -17 mg/ml for TAR$^{ES2}$ and -15 mg/ml for E-TAR$^{ES2}$ Pf1 phage (Asla biotech, Ltd.) ordering medium[46]. The RDCs used in ensemble determination were the averaged values from the two experiments and the RDC uncertainty was estimated as the RMSD between the two sets of measurements[47]. The measured RDCs of E-TAR$^{ES2}$ was scaled down by a normalization factor L$_m$ to account for differences in the degree of alignment between samples.

$$L_m = \frac{\sum_j D_j^{E-TAR^{ES2}} \times D_j^{TAR^{ES2}}}{\sum_j D_j^{TAR^{ES2}} \times D_j^{TAR^{ES2}}} \quad (1)$$

$D_j^{E-TAR^{ES2}}$ and $D_j^{TAR^{ES2}}$ are the measured RDC of the j$^{th}$ bond vector for E-TAR$^{ES2}$ and TAR$^{ES2}$. Measured RDCs are summarized in Supplementary Table 1.

**CEST experiment.** Temperature-dependent $^1$H and $^{15}$N CEST experiments were collected on wtTAR using a 600 MHz Bruker spectrometer equipped with an HCPN cryogenic probe[18,45]. Temperature-dependent $^1$H CEST experiments measured on TAR$^{ES2}$ were collected on an 800 MHz Bruker Avance TS2.1 spectrometer equipped with an HCN

cryogenic probe. Temperature-dependent $^1$H CEST experiments measured on G36U and $c^7$A 2U were collected on a 900 MHz Bruker Avance TS2.1 spectrometer equipped with an HCN cryogenic probe. Temperature-dependent $^1$H CEST experiments measured on $c^7$A wtTAR were collected on a 900 MHz Bruker Avance NEO/TS4.1 spectrometer equipped with an HCN cryogenic probe. The radiofrequency fields ($\omega_1 2\pi^{-1}$), offsets ($\Omega 2\pi^{-1}$) and mixing time ($T_{ex}$) used in the CEST experiments are listed in Supplementary Table 5. The peak intensities at each spin-lock power and offset were extracted using NMRPipe. The experimental uncertainty was obtained based on the standard deviation in peak intensities obtained from triplicate CEST experiments with zero relaxation delay for a given spin-lock power. The radiofrequency fields (RF) field inhomogeneity was measured and accounted for during CEST fitting, as previously described[45]. The exchange parameters of wtTAR, TAR$^{ES2}$, and G36U summarized in Supplementary Table 3 were obtained by fitting the normalized intensity data to a two-state Bloch-McConnell equation using an in-house Python script[18,45]. $c^7$A wtTAR and $c^7$A 2U were subjected to three-state fits with triangular topologies that simultaneously detect exchange between multiple excited states (B and C where B corresponds to ES2). The fitted parameters are summarized in Supplementary Table 4. The errors in exchange parameters were set to the standard error (SEM) derived from the square root of the diagonal elements in the covariance matrix of the fitted parameters. The $^1$H CEST profiles were also fit with and without ($p_{ES} = k_{ES} = \Delta\omega = 0$) exchange. Model selection for fits with and without exchange was performed as previously described[18] by computing Akaike (wAIC) and Bayesian information criterion (wBIC).

**Off-resonance $R_{1\rho}$ relaxation dispersion.** Off-resonance $^{15}$N $R_{1\rho}$ experiments were collected on wtTAR using a 600 MHz Bruker AVANCE-III spectrometer equipped with a triple channel cryogenic probe at 35 °C[18]. The spin-lock powers ($\omega 2\pi^{-1}$), offsets ($\Omega 2\pi^{-1}$), and delay time used in $R_{1\rho}$ experiments are listed in Supplementary Table 6. The peak intensity at each relaxation delay was extracted using NMRPipe and fitted to a monoexponential decay using an in-house python script[68]. Bloch-McConnell equations were employed to fit the off-resonance $R_{1\rho}$ value to a two-state exchange model to determine the exchange rate ($k_{ex}$), ES population ($p_{ES}$) and the difference between the ES and GS chemical shifts ($\Delta\omega = \omega_{ES} - \omega_{GS}$). The fitting errors were estimated using a Monte Carlo approach with 500 iterations[18].

## FARFAR-NMR

**Generating ensembles using FARFAR.** TAR$^{ES2}$ conformational library ($N = 100,000$) was generated using FARFAR. FARFAR is implemented as the rna_denovo program in the Rosetta Software Suite, which requires RNA sequence and optional secondary structure as input. Non-terminal residues in the lower helix (G18-C44, C19-G43, A20-U42, and G21-C41) were modeled as canonical Watson–Crick bps and imposed to a FARFAR generated idealized A form helix to reduce the run time[20]. No constraints were applied to junctional residues in and around the bulge including U23, A22-U40 which based on the U40-H3 imino proton form a labile bp, and C24-C39 for which we could not obtain any evidence for base-pairing. Pairing constraints were applied to all other bps in the upper helix including G26-C37, A27-G36, G28-A35, C29-G34, and C30-G33 for which imino resonances consistent with base pairing were observed in 2D NOESY spectra (Supplementary Fig. 8) but without specifying the bp geometry. More specific NMR derived base-pairing constraints were applied to U25-U38 (paired via the Watson-Crick face) and U51-G54(*syn*) (trans wobble) given NOE-based distance connectivity establishing the dominant paired geometry of these bps. The FARFAR input files and commands were summarized in Supplementary Table 7. The initially generated 100,000 structures were subjected to a Rosetta energy unit ≤ 0 filter to remove models that potentially have chain breaks and severe steric

clashes, after which 10,000 conformers were randomly selected to form the final conformational library. The corresponding library for E-TAR$^{ES2}$ was obtained by elongating the lower helix in the TAR$^{ES2}$ conformers by superimposing an idealized A-form helix before RDC calculation[20,48].

**RDC calculations.** RDCs were calculated using the program PALES[69] for each conformer in an ensemble. The RDC values were then averaged over all conformers in the ensemble assuming equal probability. Separate scaling factors were applied to the predicted RDCs of TAR$^{ES2}$ and E-TAR$^{ES2}$ to account for differences in the degree of alignment between samples[20].

**Sample and select (SAS).** We employed the SAS approach[41] to generate RDC-satisfying ensembles from a library. Briefly, a simulated annealing Monte Carlo sampling scheme was used to select $N$ conformers (without replacement) that minimizes the differences between the measured and predicted RDCs, in which $N$ represents the number of conformers in the ensemble or ensemble size. The effective starting temperature for simulated annealing was 100 and decreased by a factor of 0.9 in every step for a total of $5 \times 10^6$ steps. The agreement between predicted and measured RDCs is evaluated using the cost function:

$$\chi^2 = \frac{\sum_j \left( L \times D_j^{pred} - D_j^{meas} \right)^2}{N_{RDC}} \qquad (2)$$

$D_j^{pred}$ and $D_j^{meas}$ are the predicted and measured RDC of the j$^{th}$ bond vector, respectively; $L$ is the scaling factor; and $N_{RDC}$ is the total number of bond vectors. The optimal ensemble size was obtained by repeating SAS with increasing ensemble size (from $N = 1$ to 50) and finding an ensemble size at which the RDC RMSD reaches a plateau[20].

**Molecular dynamics simulations.** The FARFAR-MD library was generated by running MD simulations for each of the 10 conformers in the FARFAR-NMR ensemble as starting structures using the RNA OL3 force field[51] as recommended in the AMBER MD simulation package. Starting structures were solvated with 12 Å buffer of water[70], and were then neutralized by adding Na$^+$ ions. The equilibration phase of the simulation was performed in two steps. First 300 ns of equilibration was carried out with gradually diminishing restraints to the starting structures, allowing the system to relax and reach a stable conformation. This was followed by 600 ns of production NVT simulations using a Langevin thermostat with a collision frequency of 5 ps and a time-step of 2 fs to generate 300 snapshots per starting structure. Taken together, these simulations for all ten structures correspond to a total computational time for equilibration of ~160 h and production simulation of ~400 h on a single Titan V GPU. The final FARFAR-MD library was generated by combining the 10 starting FARFAR structures with the 3000 structures generated through MD simulation.

**Automated fragmentation quantum mechanics/molecular mechanics (AF-QM/MM) chemical shift calculations.** Automated Fragmentation quantum mechanical calculation of NMR chemical shifts (AFNMR) software[43] was used to calculate ensemble chemical shift as described previously[20]. For each RNA conformer in the ensemble, a series of five conjugate gradient energy minimization steps on heavy atoms were performed with 2 kcal/mol Å$^2$ harmonic restraints to regularize bond lengths and minimize noise in predictions. Each residue was broken into quantum mechanical fragments with a full quantum mechanical representation of heavy-atoms using a distance cutoff of 3.4 Å. The RNA atoms located outside the quantum region, water and ions present in the solvent were assigned as point charges uniformly distributed on the molecular surface. These charges

were then resolved by fitting to Poisson−Boltzmann calculations (solinprot from MEAD[71]). A local dielectric constant (ε) of 1, 4 and 80 were assigned to the quantum core, regions occupied by the conformer outside the core and the solvent, respectively. GIAO-DFT calculations in Orca5[72] (version 5.0.4) were carried out for each fragment, using the OLYP[73] functional and pcSseg-1 (triple-z plus polarization) basis set optimized for NMR shielding[74]. The predicted chemical shifts obtained from the isotropic components of the computed shielding tensor were referenced using reference shielding computations on tetramethylsilane (TMS). A linear correction was applied to the predicted chemical shifts in a nucleus type-dependent manner[20].

**Ensemble analysis.** The visualization of all ensembles was carried out using PyMOL (https://pymol.org/). All bp geometries, backbone, stacking, and sugar dihedral angles were calculated using X3DNA-DSSR[75]. The inter-helical Euler angles ($\alpha_h$, $\beta_h$, $\gamma_h$) were computed by superimposing idealized A-form geometry on three consecutive bps (lower helix: C19-G43, A20-U42, G21-C41; upper helix: G26-C37, A27-G36, G28-A35) and computing the relative orientation between these two helices[20]. Conformers with U23 flipped in or out in the FARFAR-NMR and FARFAR-MD-NMR ensembles (both $N = 10$) were identified by visual examination. For the FARFAR-MD-Library ($N = 3010$), a heavy-atom RMSD filter was used to examine the impact of excluding conformations with U23 flipped out on the RDC agreement in higher throughput. First, pairwise heavy-atom RMSD was calculated in the FARFAR-Random ensemble using the *rms2d* command in the CPPTRAJ suite[76], and the conformer with the smallest overall RMSD to remaining conformers was selected as the reference. Next, pairwise heavy-atom RMSD for the bulge motif (A22, U23, C24, C39 and U40) was calculated for all conformers ($N = 3010$) in FARFAR-MD-Library relative to the reference. Conformers with RMSD > 3.4 Å predominantly had U23 flipped out and were thus filtered out from the library.

The χ-angle of A27 and A35 in FARFAR-NMR ensemble were set to −130° using PyMOL as follow:

*cmd.set_dihedral("resi 27 and name O4'", "resi 27 and name C1'", "resi 27 and name N9", "resi 27 and name C4", −130)*

*cmd.set_dihedral("resi 35 and name O4'", "resi 35 and name C1'", "resi 35 and name N9", "resi 35 and name C4", −130)*

Conformers with steric clashes introduced by this χ-angle adjustment were identified by visual inspection and restored to their original values.

**Survey of A-G mismatches in the PDB.** All X-ray structures with a resolution of ≤3.0 Å (including unbound RNA, RNA−protein complexes and so on) were downloaded from RCSB Protein Data Bank (PDB) on July 2021 and analyzed using X3DNA-DSSR[75] to generate a JSON file library. An in-house python script was used to parse the data and create a searchable database containing RNA structural information. Tandem AG/GA bps were identified as sequentially numbered A-G mismatches in the PDB. A total of 384 tandem AG mismatches, corresponding to 36 unique bps, were identified from 117 X-ray crystal structures. Out of these, we examined 11 representative structures that corresponded to 26 unique bps (Supplementary Table 2).

**Thermodynamic analysis**
A modified van't Hoff equation was used to fit the observed temperature dependence of the forward ($k_1$) and reverse ($k_{-1}$) rate constants measured using [1]H CEST, [15]N CEST, and [15]N $R_{1\rho}$ (Supplementary Table 3). This equation accounts for statistical compensation effects and assumes a smooth energy surface[18,45,77].

$$\ln\left(\frac{k_i(T)}{T}\right) = \ln\left(\frac{k_B \kappa}{h}\right) - \frac{\Delta G_i^{\circ T}(T_{hm})}{RT_{hm}} - \frac{\Delta H_i^{\circ T}}{R}\left(\frac{1}{T} - \frac{1}{T_{hm}}\right) \quad (3)$$

where $k_i$ (i = 1, −1) is the forward and reverse rate constant computed as $k_1 = k_{ex}p_{ES}$ and $k_{-1} = k_{ex}p_{GS}$, $\Delta G_i^{\circ T}$ and $\Delta H_i^{\circ T}$ are the free energy and enthalpy of activation, $k_B$ is Boltzmann's constant, $h$ is Plank's constant, κ is the transmission coefficient (assumed to be 1), $R$ is the universal gas constant, $T$ is the temperature, and $T_{hm}$ is the harmonic mean of the experimental temperatures calculated as $T_{hm} = n/(\sum_{i=1}^{n}(1/T_i))$. The entropy of activation ($\Delta S_i^{\circ T}$) was computed using the free energy and enthalpy obtained above:

$$T_{hm}\Delta S_i^{\circ T} = \Delta H_i^{\circ T} - \Delta G_i^{\circ T}(T_{hm}) \quad (4)$$

**Reporting summary**
Further information on research design is available in the Nature Portfolio Reporting Summary linked to this article.

## Data availability
The data supporting the findings of this study are available from the corresponding authors upon request. The NMR data generated in this study are included in the published article and the Supplementary Information file and have been deposited in the BMRB database under accession code 31106 [https://doi.org/10.13018/BMR31106]. The FARFAR-MD-NMR ensemble model of TAR[ES2] ($N = 10$) used in this study is available in the PDB database under accession code 8U3M. All raw data and structural models are available on GitHub at https://github.com/alhashimilab/TAR_ES2_ensemble.

## Code availability
The Rosetta software suite is available at https://www.rosettacommons.org/software/academic. The AFNMR programs are available at https://github.com/dacase/afnmr. Custom in-house Python scripts for running sample and selection are available at https://github.com/alhashimilab/PySAS. Custom in-house Python scrips for the calculation of inter-helical Euler angles are available at https://github.com/alhashimilab/ABG_calc. Custom in-house Python scrips for the [1]H CEST data fitting and thermodynamic analysis are available at https://github.com/alhashimilab/TAR_ES2_ensemble.

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

## Acknowledgements

We thank Stephanie Gu and other members of the Al-Hashimi lab for their input. We acknowledge Duke Magnetic Resonance Spectroscopy Center and New York Structural Biology Center (NYSBC) for the technical support and resources. H.M.A. is a member of the New York Structural Biology Center (NYSBC). NMR experiments performed at NYSBC were funded by NIH grant S10 OD023499. This work was supported by US National Institute for General Medical Sciences (U54 AI150470 to H.M.A. and D.A.C and R01GM089846 to H.M.A.).

## Author contributions

A.G., L.G., and H.M.A. conceptualized the project and experimental design. A.G. with assistance from H.S. performed FARFAR calculations. A.G. performed sample and selection as well as other ensemble analysis. A.G. and L.G. prepared NMR samples. A.G., L.G. and S.P. performed NMR experiments. A.G. and H.M.A. with assistance from L.G. and R.R. analyzed the data. D.A.C. performed MD simulations and QM/MM chemical shift calculations. A.G. with assistance from R.R. prepared the figures. H.M.A. acquired funding and supervised the study. A.G. and H.M.A. wrote the manuscript with input from the remaining authors.

## Competing interests
