## [Peer Review File · Nature Communications]

An RNA excited conformational state at atomic resolutionREVIEWER COMMENTS

Reviewer #1 (Remarks to the Author):

An RNA excited ... at atomic resolution describes an experimental and computational strategy for elucidating rare states of RNA molecules. The approach is based on NMR relaxation experiments that are sensitive to excited conformers and that, therefore, provide structural constraints that allows ultimately to build a structure of the excited state through mutation. Once a suitable mutant is obtained that recapitulates excited state chemical shifts, computation can be used to generate ensembles that are consistent with RDC measures of the mutant. Overall the paper is very clear and the method powerful.

- 1) I wonder why the authors decided not to pursue a full-fledged structure determination of the mutant that mimics the excited state. Would this not be useful, even for the FARFAR-NMR approach because it would steer the choice of fragments to be those consistent with the structure. I realize that this is more work than letting a program establish potential structures based on a data-base, but given (as the authors point out) the fact that the data-base for RNA seems rather incomplete would this not be a viable approach, at least until more non-canonical base pairs populate the data-base? Or, are there so few representative fragments available in the data-base that MD is absolutely necessary? Or is it just that the NOE single structure approach would do a really poor job of recapitulating what is going on in solution where an ensemble of potentially different structures would be present?
- 2) A related question. How time consuming/difficult is it to use MD to generate the improved structures? And is it not important to validate what the MD is giving in terms of using more conventional NMR structural studies of the stabilized excited states. I am not suggesting doing this, but I think that the reader would appreciate why this particular strategy was chosen, when there are a number of paths that could have been taken and what the precise advantages are in the method described.

Overall, this work is a very important contribution and it is clear that a robust, easy to use protocol for obtaining these obviously important ensembles will significantly impact biology. I think that the strategy to go after the ES as an approach for regulating function is really excellent!

Reviewer #2 (Remarks to the Author):

The authors describe a procedure to determine structures of lowly populated conformations of RNA. The use site-directed mutagenesis targeted to stabilize these minor conformations of wild-type RNA sequences arguing that the minor conformation differs from the wild-type structure by alternate base pairing induce by register shift. The authors use chemical shift information from relaxation dispersion experiments to design the mutant structure, predict structures using FARFAR and cross-validate these models by experimental long-range residual dipolar coupling data.

At this stage of this investigation, I cannot recommend publication of the manuscript in Nat. Comm. for the following reasons.

- 1.) The authors use their most preferred, small model system TAR as only test case to introduce a new method. The validity of the approach to be generally applicable needs to be documented for a number of different RNA systems of different size and different levels of structural dynamics in addition to a single bulge.
- 2.) The use of FARFAR is not really cross-validated rigorously. Such cross-validation needs to be done including strange non-regular base pairs that can be exactly be the cause for stabilization of excited states. These are, however, not well predicted by FARFAR.

Reviewer #3 (Remarks to the Author):

In this manuscript the authors use NMR, computational ensembles, and MD simulations to determine structures of excited conformational states of an RNA molecule. They use HIV TAR as proof-of-principle. The technique involves trapping the excited state by mutagenesis, whose design is based on analysis of WT spectra. The idea is to then use dipolar coupling data to select for conformers derived by FARFAR-NMR. In the case of the TAR, ES2 excited state this proved challenging and further MD simulations were required to obtain a good fit and then 3D structure. Overall, this is a solid piece of work that adds to the arsenal of techniques for determining structures.

1)One obvious note that is missing is perhaps an explanation of how this technique compares to/ or is advantageous to traditional methods of using assignments, distance restraints, and RDCs to solve structures, especially given that the first step involves making a mutant that predominantly favors the excited state conformation.

3)While analysis of temperature-dependent exchange parameters on WT TAR to get the thermodynamic parameters is very sound, equivalent experiments on TAR ES2 construct are perhaps problematic given that it is stabilized by the UNCG type tetraloop. The authors need to elaborate on whether this would come into play.

4)The discussion veers towards the functional significance of ES2, which is contentious at best. It is not surprising that the mutation that favors ES2 is detrimental for transactivation, and the authors explanation for why it is not compatible for Tat and cyclin interaction makes sense. However, one would need a mutation that abolishes formation of ES2 to see if it is indeed important for any function at all

5)"Testing increasingly larger ensemble sizes (N),"
The value for N should be clearly stated in the main write up.

6)"Thus, ES2 could potentially function as a switch modulating the binding affinity of TAR to the Tat-SEC complex, possibly in a manner coupled to retroviral genome dimerization."
This is confusing and there is no reference. Does dimerization occur in the cytoplasm? How is it then coupled to transcriptional regulation in the nucleus?

7)The "outlier A-C1' chemical shifts" are hard to read in Fig 3.d

Reviewer #1 (Remarks to the Author):

An RNA excited ... at atomic resolution describes an experimental and computational strategy for elucidating rare states of RNA molecules. The approach is based on NMR relaxation experiments that are sensitive to excited conformers and that, therefore, provide structural constraints that allows ultimately to build a structure of the excited state through mutation. Once a suitable mutant is obtained that recapitulates excited state chemical shifts, computation can be used to generate ensembles that are consistent with RDC measures of the mutant. Overall the paper is very clear and the method powerful.

1. I wonder why the authors decided not to pursue a full-fledged structure determination of the mutant that mimics the excited state. Would this not be useful, even for the FARFAR-NMR approach because it would steer the choice of fragments to be those consistent with the structure. I realize that this is more work than letting a program establish potential structures based on a data-base, but given (as the authors point out) the fact that the data-base for RNA seems rather incomplete would this not be a viable approach, at least until more non-canonical base pairs populate the data-base? Or, are there so few representative fragments available in the data-base that MD is absolutely necessary? Or is it just that the NOE single structure approach would do a really poor job of recapitulating what is going on in solution where an ensemble of potentially different structures would be present?

This is a good point, which was also raised by reviewer 3. As noted by the reviewer, one of the concerns with pursuing conventional structure-determination is how to deal with ensemble averaging of the measured NMR parameters. Conventional structure determination protocols try to identify a single ($N=1$) structural model satisfying the ensemble-averaged NMR data. In prior work, we showed that forcing a single DNA or RNA conformation to satisfy ensemble-averaged NMR data leads to poor cross-validation with chemical shifts relative to protocols which solve for ensembles of conformations^{1,2}. Thus, ensembles were needed to account for the motional averaging of the chemical shifts. Having said that, we agree with the reviewer that, particularly in applications involving excited states, which may have unusual motifs that are underrepresented in the PDB, it may be helpful and, in some cases, even necessary to pursue full-fledged structure determination and then use these starting structures as a basis for generating ensembles. Alternatively, multi-conformer refinement approaches could also be used to determine ensembles for the ES-stabilizing mutants. To emphasize this point, we added the following paragraph to the discussion section on page 14:

“Our approach for determining conformational ensembles of the ES-stabilizing mutant relies on using state-of-the-art modeling tools including FARFAR and MD simulations to generate an initial library of conformers then using the RDCs to select conformations in the ensemble and chemical shifts to test the ensembles. Compared to conventional structure determination protocols, this approach treats ensemble averaging of the NMR data, uses physical models to address the inherent degeneracies when solving ensembles; and has the advantage of testing state-of-the-art models of RNA structure guiding their future development. In particular, in our prior work on the TAR GS ensemble³, we showed that FARFAR does a superior job sampling sugar pucker conformations relative to MD simulations whereas the current work highlights the advantages of using MD in modeling tandem G-A mismatches which may underrepresented in the PDB. However, because RNA ESs such as ES2 are likely to have unusual motifs which are underrepresented in the PDB, it may be helpful and, in some cases, even necessary to pursue full-fledged structure determination and to use the resulting structures as starting points for generating ensembles. Alternatively, multi-conformer refinement approaches could also be used to determine ensembles for the ES-stabilizing mutants^{4,5}.”

Columbia University Irving Medical Center

701 W 168th St New York, NY 10032

2. A related question. How time consuming/difficult is it to use MD to generate the improved structures? And is it not important to validate what the MD is giving in terms of using more conventional NMR structural studies of the stabilized excited states. I am not suggesting doing this, but I think that the reader would appreciate why this particular strategy was chosen, when there are a number of paths that could have been taken and what the precise advantages are in the method described.

Initiating MD simulations is straightforward given a starting 3D structure. The 250 ns equilibration for each ES2 structure required ~ 16 hours on a Titan V GPU so that for 10 structures this consumed ~7 days on a single GPU. The 600 ns production simulations for all ten structures took 17 days.

In the revised manuscript, we added these details under the methods section of “Molecular dynamics simulations”:

“Taken together, these simulations for all ten structures correspond to a total computational time for equilibration of ~ 160 hours and production simulation of ~ 400 hours on a single Titan V GPU.”

Reviewer #2 (Remarks to the Author):

The authors describe a procedure to determine structures of lowly populated conformations of RNA. The use site-directed mutagenesis targeted to stabilize these minor conformations of wild-type RNA sequences arguing that the minor conformation differs from the wild-type structure by alternate base pairing induce by register shift. The authors use chemical shift information from relaxation dispersion experiments to design the mutant structure, predict structures using FARFAR and cross-validate these models by experimental long-range residual dipolar coupling data.

At this stage of this investigation, I cannot recommend publication of the manuscript in Nat. Comm. for the following reasons.

1. The authors use their most preferred, small model system TAR as only test case to introduce a new method. The validity of the approach to be generally applicable needs to be documented for a number of different RNA systems of different size and different levels of structural dynamics in addition to a single bulge.

We understand the point expressed by the reviewer i.e. a new method should ultimately be demonstrated on several systems to establish generality. As noted in the original submission, we and others have used our chemical-shift strategy to determine the secondary structures of RNA excited conformational states and identify tentative mutants stabilizing the excited state in a variety of systems, including the ribosomal A-site⁶, HIV-1 stem-loop¹⁶ and rev response element⁷, the P5abc domain from 'Tetrahymena' group I intron ribozyme⁸, microRNA-mRNA complex of miR-34a targeting Sirt1 mRNA⁹, fluoride riboswitch aptamer¹⁰, and microRNA-21¹¹. While the validity of these mutants as ES-mimics requires more extensive measurements of relaxation dispersion data, as was done here for TAR ES2, these studies collectively support the general applicability of the approach, especially given that the identification of ES-mutants is the most challenging step in our pipeline. We have emphasized this more strongly in the revised manuscript as well as the importance of establishing the generality of the approach in future studies in the discussion on page 13:

“The validity of these mutants as ES-mimics could be further verified in the future through extensive measurements of relaxation dispersion data across various nuclei and residues, and this in turn, could provide a route for solving the 3D structures of these other RNA ESs. These applications are needed to test the general applicability of the approach on diverse RNAs.”

However, we would like to respectfully push back on the request to demonstrate the approach on other systems in the current manuscript. Our study reports a new approach which allowed us to determine the first high-resolution structure of a fleeting short-lived RNA ES. Historically, in structural biology, such milestones are often achieved on one system. This is because significant time and resources need to be invested in developing and rigorously testing core aspects of the new technology. Indeed, all six prior papers reporting 3D structures of protein excited states, three of which were published in *Nature* or *Science* (summarized in Table 1 below), developed and applied a new method on one system. The structure of the TAR ES2 is the culmination of more than a decade of work requiring the development and application of experiments for measuring ¹H, ¹³C, and ¹⁵N relaxation dispersion / CEST data as well as multi-RDC data sets. We therefore would like to avoid breaking with precedent and delaying publication of our work potentially several years to apply the method on a different system.

Table 1. Summary of existing protein excited states ensembles

System	Type	No. of ES structures	Additional system(s) tested	Reference
--------	------	----------------------	-----------------------------	-----------

Abp1p SH3 domain	Protein	1	None	Vallurupalli, P., ..., Kay, L.E. et al. Proc Natl Acad Sci U S A 105, 11766-71 (2008).
FF domain	Protein	1	None	Korzhev, D.M., ..., Kay, L.E. et al. Science 329, 1312-6 (2010).
L24A-FF domain	Protein	1	None	Korzhev, D.M., ..., Kay, L.E. et al. J Am Chem Soc 133, 10974-10982 (2011).
Fyn SH3 domain	Protein	1	None	Neudecker, P., ..., Kay, L.E. et al. Science 336, 362-366 (2012).
T4 lysozyme	Protein	1	Compared simulated MD trajectories on two mutants of the same system	Vallurupalli, P., ..., Kay, L.E. et al. Chemical science 7, 3602-3613 (2016).
Adenylate kinase	Protein	1	Shows feasibility on two more systems using simulated data (Calmodulin and Src kinase)	Stiller, J.B., ..., Kern, D. et al. Nature 603, 528-535 (2022).

2. The use of FARFAR is not really cross-validated rigorously. Such cross-validation needs to be done including strange non-regular base pairs that can be exactly be the cause for stabilization of excited states. These are, however, not well predicted by FARFAR.

The reviewer may have misunderstood what was done in our study. In their remarks, the reviewer noted that in our manuscript, we “predict structures using FARFAR and cross-validate these models by experimental long-range residual dipolar coupling data.” This is incorrect. Rather, the ensemble was determined using ensemble-averaged residual dipolar couplings (RDCs) to guide the selection of conformers from a FARFAR-generated conformational library. The RDC-derived ensemble was then cross-validated using orthogonal ¹H, ¹³C, and ¹⁵N chemical shift data, which were not used in the ensemble determination. To clarify these crucial points, we changed one of the sub-headings to make clear that RDCs are used in the ensemble determination on page 8,

“Determining the ES2 conformational ensemble using FARFAR and RDCs”

In addition, we added the following opening sentence on page 8:

“We used FARFAR-NMR to determine ensembles of the ES-mutant by integrating FARFAR structure prediction with NMR RDC data and then used chemical shifts to cross-validate the generated ensemble.”

Cross-validating RNA ensembles using rich and orthogonal chemical shift data spanning the base and sugar moieties is a proven method which we have developed and applied over many years^{3,12,13}. But even more compelling is that the cross-validation with the chemical shifts achieved its intended purpose, revealing that the G-A mismatches were indeed inadequately modeled in the original FARFAR-NMR ensemble, and this is what prompted us to further refine the ensemble using MD. In other words, our initial ensemble failed our rigorous chemical shift cross-validation tests leading us to optimize the ensemble using MD.

Nevertheless, to address any concerns about the accuracy of the ensemble, we performed additional experiments and incorporated new data in the revised manuscript to evaluate key predictions made by our ensemble concerning the conformation of the A-G mismatches referred to by the reviewer.

It is well established that the A-G mismatch can adopt a variety of conformations, including $A_{anti}\text{-}G_{anti}$, two alternative Hoogsteen conformations in which one of the purine bases flips into the *syn* conformation ($A_{syn}\text{-}G_{anti}$ and $A^+_{anti}\text{-}G_{syn}$), and the sheared conformation in which the two bases are shifted relative to each other in a *trans* configuration with hydrogen bonds formed between the Hoogsteen face of the adenine and the sugar face of guanine. According to our FARFAR-NMR and optimized FARFAR-MD-NMR ensemble, both A-G mismatches in the tandem AG mismatches adopt the $A_{anti}\text{-}G_{anti}$ conformation. To further cross-validate this A-G conformation in the ensemble, we performed experiments in which we substituted A27 in wtTAR with its isosteric base analog 7-deaza-adenosine (c^7A), replacing N7 with C7H7. Based on our ensembles and the $A_{anti}\text{-}G_{anti}$ conformation, the modification should not impact the chemical exchange between the GS and ES2. On the other hand, the modification is predicted to substantially destabilize ES2 and quench the chemical exchange if the A-G mismatch adopted either the $A_{syn}\text{-}G_{anti}$ Hoogsteen or sheared conformation¹⁴. We used chemical synthesis to prepare wtTAR in which A27 was substituted with c^7A . Indeed, based on the ¹H CEST profiles, the c^7A modification did not quench the exchange with ES2; on the contrary, it increased the population of the ES2 by 8-fold. Similar results were obtained with the 2-bulge variant of TAR (2U), which also forms ES2¹⁵. The increase in the ES2 population is most likely due to the destabilization of the GS. Therefore, these data further support the validity of the $A_{anti}\text{-}G_{anti}$ conformation in the FARFAR-MD-NMR ensemble.

These new data are now included under a new subsection called “*Cross-validating the ensemble using single-atom substitutions*” on page 11 and a new Figure 4 and Supplementary Fig. 10.

“Based on our FARFAR-NMR and FARFAR-MD-NMR ensembles, the tandem AG mismatches adopt the $A_{anti}\text{-}G_{anti}$ conformation, not other commonly observed conformations such as the $A_{syn}\text{-}G_{anti}$ Hoogsteen and sheared A-G (Fig. 4a). Thus, our ensemble predicts that replacing the adenosine base with its isosteric base analog 7-deaza-adenosine (c^7A) that replaces N7 by C7H7 should not impact the formation of the ES2 (Fig. 4a); on the other hand, based on our prior work on DNA¹⁴, the modification should destabilize and potentially quench exchange with ES2 if the AG mismatch in ES2 adopted either the $A_{syn}\text{-}G_{anti}$ Hoogsteen or sheared A-G conformation (Fig. 4a). We tested this prediction using chemical synthesis to prepare wtTAR in which A27 is substituted with c^7A (Fig. 4b). The G28-H1 and G26-H1 ¹H CEST profiles could be combined in a global fit, yielding downfield shifted G28-H1 ($\Delta\omega$ of ~ -0.7 ppm) and upfield shifted G26-H1 ($\Delta\omega$ of ~ 0.5 ppm) chemical shifts consistent with the Watson-Crick $A_{anti}\text{-}G_{anti}$ bps in the wtTAR ES2 (Fig. 4c, d and Supplementary Fig. 10a, c). The exchange rate of $k_{ex} = 614 \pm 52$ s⁻¹ was in very good agreement with values measured for the wtTAR ES2 ($k_{ex} = 737 \pm 39$ s⁻¹) using ¹H CEST (Fig. 4d). Indeed, the modification did not quench the exchange rather it increased the population of the ES2 by ~ 8 -fold, possibly due to the destabilization of the GS¹⁴. Similar results were obtained robustly for the HIV-2 TAR variant (2U) with the UU dinucleotide bulge (Supplementary Fig. 10b, d)”

Figure 4. Cross-validating the $A_{27}^{anti}-G_{36}^{anti}$ conformation in ES2 using single-atom substitutions. (a) The c^7A substitution of A27 is predicted to selectively destabilize $A_{syn}-G_{anti}$ Hoogsteen and sheared A-G conformations but not the $A_{anti}-G_{anti}$ conformation formed in the TAR^{ES2} ensemble. (b) c^7A wtTAR preserves the dynamic equilibrium between the GS and ES2. The modified residues are highlighted in red, and residues showing exchange contributions to ES2 in the 1H CEST experiment are circled in pink. (c) Representative 1H CEST profile for G28-H1 showing an exchange contribution. RF field powers used are color-coded. Remaining data are shown in Supplementary Fig. 10. (d) Comparison of the kinetic exchange rate ($k_{ex} = k_1 + k_{-1}$), the population (p_{ES2}) and chemical shift difference ($\Delta\omega = \omega_{ES2} - \omega_{GS}$) measured on wtTAR and c^7A wtTAR. The error bars denote the fitting errors from the global fit.

Supplementary Figure 10. c^7A preserves the GS to ES2 exchange dynamic in different sequence contexts. (a-b) Comparison of 1H 1D spectra of the imino region of (a) wtTAR (green) and c^7A wtTAR (blue), and (b) 2U (orange) and c^7A 2U (brown); the modified residues are highlighted in red, and residues showing exchange contributions to ES2 in the 1H

CEST experiment are circled in pink. The absence of U38-H3 peaks is in line with prior studies in DNA in which significant line broadening was observed at c⁷A modified and neighboring base pairs¹⁴. (c-d) ¹H CEST profiles measured in (c) c⁷A wtTAR and (d) c⁷A 2U at 25°C. A new excited state with $\Delta\omega$ of ~ -2.6 ppm was observed at G28 in the tandem AG mismatch, which could be attributed to a protonated A_{syn}⁺-G_{anti} conformation. ¹H CEST data measured on G28 were fit to a three-state model with (+ex) or without (-ex, $k_{ex} = \Delta\omega = p_{ES} = 0$) exchange using Bloch–McConnell equations. The G26 ¹H CEST profile was fitted to a two-state model in c⁷A wtTAR and a three-state model in c⁷A 2U. Model selection (+ex or -ex) was determined based on the reduced chi-square ($r\chi^2$), Akaike’s (wAIC), and Bayesian (wBIC) information criterion weights (Methods). Also shown are corresponding residual plots (normalized experimental intensity - fitted normalized intensity). The error bars for the ¹H CEST profile are smaller than the data point and were derived from the standard deviation of three measurements of peak intensity with zero relaxation delay. RF field powers used are color-coded.

To incorporate these new data, we updated the relevant methods sections “RNA Preparation” to include solid-phase synthesis of c⁷A modified samples as follows: “... c⁷A wtTAR and c⁷A 2U RNA were synthesized using a MerMade 6 Oligo Synthesize ...”. We also updated the “CEST experiment” methods section to include: “Temperature-dependent ¹H CEST experiments measured on G36U and c⁷A 2U were collected on a 900 MHz Bruker Avance TS2.1 spectrometer equipped with an HCN cryogenic probe. Temperature-dependent ¹H CEST experiments measured on c⁷A wtTAR were collected on a 900 MHz Bruker Avance NEO/TS4.1 spectrometer equipped with an HCN cryogenic probe ... c⁷A wtTAR and c⁷A 2U were subjected to three-state fits with triangular topologies that simultaneously detect exchange between multiple excited states (B and C where B corresponds to ES2). The fitted parameters are summarized in Supplementary Table 4.”

In addition, we added a new Supplementary Table 4 and updated Supplementary Table 5 detailing ¹H CEST results and parameters related to the c⁷A wtTAR and c⁷A 2U:

Supplementary Table 4. Exchange parameters obtained from 3-state fitting of ¹H CEST data for c⁷A wtTAR and c⁷A 2U.

Sample	c ⁷ A wtTAR 25°C, pH 6.4, T _{ex} = 80ms				c ⁷ A 2U 25°C, pH 6.4, T _{ex} = 80ms			
	Shared 3-state Fitting		Individual Fitting		Shared 3-state Fitting		Individual Fitting	
Resonance	G28	G26	G28	G26	G28	G26	G28	G26
p _B (%)	1.69±0.1		1.73±0.08	1.68±1.06	1.52±0.17		1.18±0.46	1.94±0.28
p _C (%)	0.29±0.02	N/A	0.29±0.02	N/A	0.19±0.06	0.53±0.08	0.20±0.09	0.54±0.10
Δω _B (ppm)	-0.68±0.01	0.51±0.01	-0.68±0.01	0.52±0.01	-0.50±0.04	0.40±0.02	-0.31±0.17	0.38±0.02
Δω _C (ppm)	-2.57±0.03	N/A	-2.58±0.02	N/A	-2.57±0.09	-2.28±0.33	-2.24±0.26	-2.27±0.36
k _{exAB} (s ⁻¹)	614±52		657±48	556±116	928±177		757±609	1152±271
k _{exAC} (s ⁻¹)	149±192	N/A	1.0±142	N/A	1.7±557	13884±3155	45±2781	13703±3457
k _{exBC} (s ⁻¹)	1908±305	N/A	2074±237	N/A	2294±1093	1.1±522	10991±4487	1±594
R ₁ (s ⁻¹)	11.27±0.02	20.06±0.02	11.27±0.01	20.05±0.02	10.39±0.01	14.45±0.02	10.39±0.01	14.46±0.02
R ₂ (s ⁻¹)	40.84±0.48	56.53±0.65	40.77±0.37	56.96±0.82	34.97±0.51	33.43±5.09	33.94±1.07	30.14±5.52
Red. rχ ²	4.15		2.42	5.89	20.06		13.75	26.5

Supplementary Table 5. List of RF powers ($\omega_1 2\pi^{-1}(s^{-1})$) and offsets ($\Omega 2\pi^{-1}(s^{-1})$) used in the ¹H CEST experiments for c⁷A wtTAR and c⁷A 2U.

T (°C)	$\omega_1 2\pi^{-1}(s^{-1})$	$\Omega 2\pi^{-1}(s^{-1})$
¹ H CEST, c ⁷ A wtTAR (pH 6.4, 90% H ₂ O:10% D ₂ O)		
25 °C, T _{ex} = 80 ms	10	[-5391, -4941, -4491, -4041, -3591, -3483, -3375, -3267, -3159, -3051, -2943, -2835, -2727, -2619, -2511, -2403, -2295, -2187, -2079, -1971, -1863, -1755, -1647, -1539, -1431, -1323, -1215, -1107, -999, -891, -785, -679, -573, -467, -362, -256, -150, 61, 167, 273, 379, 485,
	50	
	100	

	200	590, 696, 802, 908, 1016, 1124, 1232, 1340, 1448, 1556, 1664, 1772, 1880, 1988, 2096, 2204, 2312, 2420, 2528, 2636, 2744, 2852, 2960, 3068, 3176, 3284, 3392, 3500, 3608, 4058, 4508, 4958, 5408]
	500	
	1000	
	4000	
¹ H CEST, c ⁷ A 2U (pH 6.4, 90% H ₂ O:10% D ₂ O)		
25 °C, T _{ex} = 80 ms	10	[-6639, -6189, -5739, -5288, -4838, -4388, -3938, -3830, -3722, -3614, -3506, -3398, -3290, -3182, -3074, -2966, -2858, -2750, -2642, -2534, -2426, -2318, -2210, -2102, -1994, -1886, -1778, -1670, -1562, -1454, -1346, -1238, -1132, -1027, -921, -815, -709, -603, -497, -391, -285, -179, -74, 31, 137, 243, 349, 455, 561, 669, 777, 885, 993, 1101, 1209, 1317, 1425, 1533, 1641, 1749, 1857, 1965, 2073, 2181, 2289, 2397, 2505, 2613, 2721, 2829, 2937, 3045, 3153, 3261, 3711, 4161, 4611, 5061, 5511, 5961]
	50	
	100	
	200	
	500	
	1000	
	4000	

Reviewer #3 (Remarks to the Author):

In this manuscript the authors use NMR, computational ensembles, and MD simulations to determine structures of excited conformational states of an RNA molecule. They use HIV TAR as proof-of-principle. The technique involves trapping the excited state by mutagenesis, whose design is based on analysis of WT spectra. The idea is to then use dipolar coupling data to select for conformers derived by FARFAR-NMR. In the case of the TAR, ES2 excited state this proved challenging and further MD simulations were required to obtain a good fit and then 3D structure. Overall, this is a solid piece of work that adds to the arsenal of techniques for determining structures.

1. One obvious note that is missing is perhaps an explanation of how this technique compares to/ or is advantageous to traditional methods of using assignments, distance restraints, and RDCs to solve structures, especially given that the first step involves making a mutant that predominantly favors the excited state conformation.

We thank the reviewer for raising this point. We addressed this point in our responses to Reviewer 1.

2. While analysis of temperature-dependent exchange parameters on WT TAR to get the thermodynamic parameters is very sound, equivalent experiments on TAR ES2 construct are perhaps problematic given that it is stabilized by the UNCG type tetraloop. The authors need to elaborate on whether this would come into play.

We measured the temperature dependence of the exchange kinetics of the TAR ES2 construct to verify the results we obtained on the TAR GS construct. Nevertheless, we agree with the reviewer that this analysis is potentially confounded by the replacement of wild-type (WT) apical loop with the UNCG tetraloop. To address this concern, we performed chemical exchange measurements on a different ES2-mutant, which preserves the WT apical loop and replaces a helical residue G36 with U36. The G36U ES2-mutant¹⁶ replaces energetically unfavorable mismatches in the ES2 with Watson-Crick base pairs while replacing a Watson-Crick base pair in the GS with a mismatch. We previously showed that the G36U mutant also adopts the ES2 conformation as the dominant conformation and inhibits cellular transactivation by preferentially stabilizing the inactive ES2 conformation¹⁶. We prepared the G36U ES2-mutant and measured temperature-dependent ¹H CEST profiles. We find that the G36U mutant back-exchanges with a GS-like conformation with equilibrium population of $\sim 3.7 \pm 2.3\%$. Based on the temperature dependence of the exchange kinetics of the G36U TAR mutant, we determined an oppositely signed $\Delta H_{GS-ES2} = 16.5 \pm 2.3$ kcal/mol relative to WT TAR, providing additional support that ES2 is enthalpically favored but entropically disfavored relative to the GS.

These new data are included in the section entitled “*ES2 is entropically disfavored relative to the GS*” as follows: “*In addition, analogous experiments on the TAR^{ES2} and G36U mutants (Supplementary Fig. 12-14), both of which back exchange with a GS-like conformation¹⁷, yielded oppositely signed ΔH_{GS-ES2} .*”

The data are shown in Supplementary Figure 14:

Supplementary Figure 14. Kinetic-thermodynamic analysis of the excited-to-ground state transition. (a) Chemical exchange between the GS and ES in the G36U mutant. Also shown are the populations (p_{GS} and p_{ES2}) and exchange rate constant deduced using 1H CEST experiment at 25°C. The mutation site is colored in red. (b) Overlay of 1D 1H imino spectra measured for G36U and TAR^{ES2} , showing similar 1H chemical shifts. (c) Temperature-dependent G36U exchange profiles. 1H CEST profiles measured on U38-H3 as a function of temperature (15°C, 20°C, and 25°C). 1H CEST data were fit to a two-state model with (+ex) or without (-ex) exchange using Bloch–McConnell equations. Model selection (+ex or -ex) was determined based on χ^2 , Akaike’s (wAIC), and Bayesian (wBIC) information criterion weights (Methods). Also shown are corresponding residual plots (normalized experimental intensity - fitted normalized intensity). The error bars for the 1H CEST profile are smaller than the data point and were derived from the standard deviation of three measurements of peak intensity with zero relaxation delay. (d) Comparison of the 1H chemical shifts difference ($\Delta\omega = \omega_{ES} - \omega_{GS}$) measured on wtTAR (green), TAR^{ES2} (pink), and G36U (blue). The open circle denotes the absolute value of wtTAR $\Delta\omega$. In all cases, the uncertainty in $\Delta\omega < 0.01$ ppm. (e) Temperature-dependent forward (k_1) and reverse (k_{-1}) rate constants for the ES2 to GS exchange obtained using 1H CEST in G36U. (f) The kinetic-thermodynamic profile for exchange between the ES2 and GS in G36U via a transition state (\ddagger), showing activation and net free energy (G), enthalpy (H), and entropy (TS) changes with ES2 referenced to 0.

To incorporate these new data, we updated the relevant methods sections “RNA Preparation” to include solid-phase synthesis of G36U mutant as follows: “... G36U mutant ... were synthesized using a MerMade 6 Oligo Synthesize ...”. We also updated the “CEST experiment” methods section to include: “The exchange parameters of wtTAR, TAR^{ES2} , and G36U summarized in Supplementary Table 3 were obtained by fitting the normalized intensity data to a two-state Bloch-McConnell equation using an in-house Python script ...”

In addition, we updated Supplementary Tables 3 and 5 to include the new data measured for the G36U mutant:

Supplementary Table 3. Exchange parameters obtained from 2-state fitting of 1H CEST data for G36U.

Sample	Resonance	p_{ES} (%)	k_{ex} (s^{-1})	$\Delta\omega$ (ppm)	R_1 (s^{-1})	R_2 (s^{-1})	Red. χ^2
G36U 15°C, pH 6.4, $T_{ex} = 80$ ms	U38-H3	1.59 ± 0.43	221.82 ± 40.82	3.03 ± 0.01	7.17 ± 0.01	39.32 ± 0.31	4.89
G36U 20°C, pH 6.4, $T_{ex} = 80$ ms	U38-H3	2.52 ± 0.69	462.1 ± 19.66	3.04 ± 0.00	10.5 ± 0.01	40.79 ± 0.68	5.10
G36U 25°C, pH 6.4, $T_{ex} = 100$ ms	U38-H3	3.67 ± 2.34	853.4 ± 71.1	3.04 ± 0.01	16.04 ± 0.03	46.01 ± 2.65	3.43

Supplementary Table 5. List of RF powers ($\omega_1 2\pi^{-1}(s^{-1})$) and offsets ($\Omega 2\pi^{-1}(s^{-1})$) used in the 1H CEST experiments for G36U.

T (°C)	$\omega_1 2\pi^{-1}(s^{-1})$	$\Omega 2\pi^{-1}(s^{-1})$
1H CEST, G36U (pH 6.4, 90% H₂O:10% D₂O)		
15 °C, $T_{ex} = 80$ ms	10	[-5391, -4941, -4491, -4041, -3591, -3483, -3375, -3267, -3159, -3051, -2943, -2835, -2727, -2619, -2511, -2403, -2295, -2187, -2079, -1971, -1863, -1755, -1647, -1539, -1431, -1323, -1215, -1107, -999, -891, -785, -679, -573, -467, -362, -256, -150, 61, 167, 273, 379, 485, 590, 696, 802, 908, 1016, 1124, 1232, 1340, 1448, 1556, 1664, 1772, 1880, 1988, 2096, 2204, 2312, 2420, 2528, 2636, 2744, 2852, 2960, 3068, 3176, 3284, 3392, 3500, 3608, 4058, 4508, 4958, 5408]
	50	
	100	
	500	
20 °C, $T_{ex} = 80$ ms	10	[-5376, -4926, -4476, -4026, -3576, -3468, -3360, -3252, -3144, -3036, -2928, -2820, -2712, -2604, -2496, -2388, -2280, -2172, -2064, -1956, -1848, -1740, -1632, -1524, -1416, -1308, -1200, -1092, -984, -876, -770, -665, -559, -453, -347, -241, -135, -29, 76, 182, 287, 393, 499, 605, 711, 817, 923, 1031, 1139, 1247, 1355, 1463, 1571, 1679, 1787, 1895, 2003, 2111, 2219, 2327, 2435, 2543, 2651, 2759, 2867, 2975, 3083, 3191, 3299, 3407, 3515, 3623, 4073, 4523, 4973, 5423]
	50	
	100	
	500	
25 °C, $T_{ex} = 100$ ms	10	[-5388, -4938, -4488, -4038, -3588, -3480, -3372, -3264, -3156, -3048, -2940, -2832, -2724, -2616, -2508, -2400, -2292, -2184, -2076, -1968, -1860, -1752, -1644, -1536, -1428, -1320, -1212, -1104, -996, -888, -782, -676, -570, -464, -358, -253, -41, 64, 276, 382, 488, 593, 699, 805, 911, 1019, 1127, 1235, 1343, 1451, 1559, 1667, 1775, 1883, 1991, 2099, 2207, 2315, 2423, 2531, 2639, 2747, 2855, 2963, 3071, 3179, 3287, 3395, 3503, 3611, 4061, 4511, 4961, 5411]
	50	
	100	
	200	
	500	

3. The discussion veers towards the functional significance of ES2, which is contentious at best. It is not surprising that the mutation that favors ES2 is detrimental for transactivation, and the authors explanation for why it is not compatible for Tat and cyclin interaction makes sense. However, one would need a mutation that abolishes formation of ES2 to see if it is indeed important for any function at all

We agree with the reviewer that mutants abolishing ES2 while preserving the WT conformation would provide a unique opportunity to study what, if any, are the functional roles of ES2. Unfortunately, because of the duplicated copies of TAR in the 5' and 3' LTRs, it is technically difficult to mutate TAR and broadly assess the consequence on viral replication. However, in a prior paper¹⁶, we showed that mutants predicted to destabilize ES2 have little effect on Tat-dependent cellular transactivation. These results were not surprising because Tat-dependent cellular transactivation is a structurally well-characterized process in which the Tat-SEC complex binds to the ground-state TAR secondary structure in a base-triple conformation. However, these assays did not test the potential role of the TAR ES2 in genome dimerization or in releasing the Tat-SEC complex.

As noted in the original submission, we speculate that ES2 could play a role in the dimerization and packaging of the retroviral genome. It has been shown that TAR is required for proper genome dimerization and/or packaging through mechanisms that are not fully understood¹⁸⁻²⁰. In addition, we previously showed that the TAR ES2 has a high propensity to form kissing dimers^{15,16}. Finally, deletion of the UCU trinucleotide bulge, which inhibits the formation of ES2²¹, significantly impairs retroviral genome dimerization²². We plan on pursuing studies examining the role of ES2 in dimerization and packaging.

To emphasize these points, we updated the introduction paragraph speculating on the potential functional roles of ES2 on page 4,

“While no functional role has yet been assigned to the TAR ES2, point-substitution mutations making ES2 the dominant conformation promote kissing-loop dimerization¹⁵, hinting to a potential role in

genome dimerization and packaging^{18-20,22} as well as potentially inhibit cellular transactivation possibly pointing to a role in releasing Tat-SEC complex¹⁶. Regardless of its potential functional roles, the 3D structure of the ES2 is of great interest for the design of anti-HIV therapeutics, which inhibit transcriptional activation by preferentially binding and stabilizing this alternative inactive TAR conformation^{16,17}.”

We also included a new paragraph in the discussion on page 16:

“While no functional role has been assigned to the TAR ES2, we speculate that ES2 could play a role in the dimerization and packaging of the retroviral genome. It has been shown that TAR is required for proper genome dimerization and/or packaging through mechanisms that are not fully understood¹⁸⁻²⁰. In addition, we previously showed that the TAR ES2 has a high propensity to form kissing dimers^{15,16}. Finally, deletion of the UCU trinucleotide bulge, which inhibits the formation of ES2²¹, significantly impairs retroviral genome dimerization²². These functional roles can be tested in the future by examining the consequence of introducing TAR ES2-stabilizing and ES2-destabilizing mutations in dimerization and packaging assays.”

4. “Testing increasingly larger ensemble sizes (N),” the value for N should be clearly stated in the main write up.

We agree with the reviewer and have updated the relevant statement on page 8 to improve its clarity: *“Testing increasingly larger ensemble sizes (N), starting with N = 1 up to N = 49, an optimal ensemble with N = 10 conformers (see Supplementary Fig. 4f) could be obtained, which showed improved RDC agreement across both helices and the bulge.”*

For further clarification, we also included the values for N in all RDC correlation plots.

5. “Thus, ES2 could potentially function as a switch modulating the binding affinity of TAR to the Tat-SEC complex, possibly in a manner coupled to retroviral genome dimerization.”

This is confusing and there is no reference. Does dimerization occur in the cytoplasm? How is it then coupled to transcriptional regulation in the nucleus?

We thank the reviewer for their comment and apologize for the lack of clarity. Genomic RNA dimerization occurs in the cytoplasm²³ at the plasma membrane in the presence of Gag²⁴. The changes described above on page 4 address this point along with the functional significance of ES2.

6. The “outlier A-C1' chemical shifts” are hard to read in Fig 3.d

We thank the reviewer for this comment and apologize for the confusion. The “outlier A-C1' chemical shifts” refers to the chemical shift correlation plot in Figure 3.c. Figure 3.d is the RDC correlation plot. To improve the clarity, we updated the figure legend of Figure 3 c, d: *“(c-d) Comparison between measured and ensemble predicted (c) representative chemical shifts measured on wtTAR ES2 (The outlier A-C1' chemical shifts are highlighted in open green symbols) and (d) RDCs (TAR^{ES2} + E-TAR^{ES2}).”*

Additional Changes

- We corrected a minor typo in Supplementary Figure 1 (a), which previously had incorrect values for wAIC and wBIC for residues without exchange. They have now been corrected in the revised manuscript, as shown below.

- We have included the ^1H CEST exchange parameters for wtTAR measured at 40°C which were missing in the prior Supplementary Table 3:

Sample	Resonance	p_{ES} (%)	k_{ex} (s^{-1})	$\Delta\omega$ (ppm)	R_1 (s^{-1})	R_2 (s^{-1})	Red. $r\chi^2$
wtTAR 40°C , $T_{\text{ex}} = 80\text{ms}$	U38-H3	0.24 ± 0.04	4417.05 ± 1044.8	-2.73 ± 0.17	31.09 ± 0.07	37.97 ± 1.07	6.20

- We have deposited the ensemble model presented in the manuscript to the PDB, available with PDB ID **8U3M**. Additionally, the measured RDC data and chemical shift assignments have also been deposited and are available as NIH-XPLOR and NMR-STAR file formats in the associated BMRB deposition **31106**. We have also summarized the above changes in a new section titled "Data availability": "All raw data and structural models are available on GitHub at https://github.com/alhashimilab/TAR_ES2_ensemble. The FARFAR-MD-NMR ensemble model of TAR^{ES2} ($N = 10$) along with all relevant NMR data used for RDC optimization and chemical shift cross-validation has been deposited in the PDB, available with PDB ID 8U3M and BMRB ID 31106. Any further information is available from the corresponding authors upon request."

4. We have also included a new section titled “Code availability” to summarize the accessibility of software suites and in-house python script: *“The Rosetta software suite is available at <https://www.rosettacommons.org/software/academic>. The AFNMR programs are available at <https://github.com/dacase/afnmr>. Custom in-house Python scripts for running sample and selection are available at <https://github.com/alhashimilab/PySAS>. Custom in-house Python scrips for the calculation of inter-helical Euler angles are available at https://github.com/alhashimilab/ABG_calc. Custom in-house Python scrips for the ¹H CEST data fitting and thermodynamic analysis are available at https://github.com/alhashimilab/TAR_ES2_ensemble.”*

References

1. Shi, H. et al. Atomic structures of excited state A-T Hoogsteen base pairs in duplex DNA by combining NMR relaxation dispersion, mutagenesis, and chemical shift calculations. *J. Biomol. NMR* **70**, 229-244 (2018).
2. Sathyamoorthy, B. et al. Insights into Watson–Crick/Hoogsteen breathing dynamics and damage repair from the solution structure and dynamic ensemble of DNA duplexes containing m1A. *Nucleic Acids Res.* **45**, 5586-5601 (2017).
3. Shi, H. et al. Rapid and accurate determination of atomistic RNA dynamic ensemble models using NMR and structure prediction. *Nat. Commun.* **11**, 5531 (2020).
4. Vögeli, B., Kazemi, S., Güntert, P. & Riek, R. Spatial elucidation of motion in proteins by ensemble-based structure calculation using exact NOEs. *Nat. Struct. Mol. Biol.* **19**, 1053-1057 (2012).
5. Anthis, N.J. & Clore, G.M. Visualizing transient dark states by NMR spectroscopy. *Q. Rev. Biophys.* **48**, 35-116 (2015).
6. Dethoff, E.A., Petzold, K., Chugh, J., Casiano-Negroni, A. & Al-Hashimi, H.M. Visualizing transient low-populated structures of RNA. *Nature* **491**, 724-8 (2012).
7. Chu, C.C., Plangger, R., Kreutz, C. & Al-Hashimi, H.M. Dynamic ensemble of HIV-1 RRE stem IIB reveals non-native conformations that disrupt the Rev-binding site. *Nucleic Acids Res.* **47**, 7105-7117 (2019).
8. Xue, Y., Gracia, B., Herschlag, D., Russell, R. & Al-Hashimi, H.M. Visualizing the formation of an RNA folding intermediate through a fast highly modular secondary structure switch. *Nat. Commun.* **7**, 1-11 (2016).
9. Baronti, L. et al. Base-pair conformational switch modulates miR-34a targeting of Sirt1 mRNA. *Nature* **583**, 139-144 (2020).
10. Zhao, B., Guffy, S.L., Williams, B. & Zhang, Q. An excited state underlies gene regulation of a transcriptional riboswitch. *Nat. Chem. Biol.* **13**, 968-974 (2017).
11. Baisden, J.T., Boyer, J.A., Zhao, B., Hammond, S.M. & Zhang, Q. Visualizing a protonated RNA state that modulates microRNA-21 maturation. *Nat. Chem. Biol.* **17**, 80-88 (2021).
12. Salmon, L. et al. Modulating RNA Alignment Using Directional Dynamic Kinks: Application in Determining an Atomic-Resolution Ensemble for a Hairpin using NMR Residual Dipolar Couplings. *J Am Chem Soc* **137**, 12954-12965 (2015).
13. Salmon, L., Bascom, G., Andricioaei, I. & Al-Hashimi, H.M. A General Method for Constructing Atomic-Resolution RNA Ensembles using NMR Residual Dipolar Couplings: The Basis for Interhelical Motions Revealed. *J Am Chem Soc* **135**, 5457-5466 (2013).
14. Nikolova, E.N., Gottardo, F.L. & Al-Hashimi, H.M. Probing Transient Hoogsteen Hydrogen Bonds in Canonical Duplex DNA Using NMR Relaxation Dispersion and Single-Atom Substitution. *J Am Chem Soc* **134**, 3667-3670 (2012).
15. Merriman, D.K. et al. Shortening the HIV-1 TAR RNA Bulge by a Single Nucleotide Preserves Motional Modes over a Broad Range of Time Scales. *Biochemistry* **55**, 4445-4456 (2016).
16. Ganser, L.R. et al. Probing RNA Conformational Equilibria within the Functional Cellular Context. *Cell Rep.* **30**, 2472-2480.e4 (2020).
17. Ganser, L.R., Kelly, M.L., Patwardhan, N.N., Hargrove, A.E. & Al-Hashimi, H.M. Demonstration that Small Molecules can Bind and Stabilize Low-abundance Short-lived RNA Excited Conformational States. *J. Mol. Biol.* **432**, 1297-1304 (2020).
18. Harrich, D., Hooker, C.W. & Parry, E. The human immunodeficiency virus type 1 TAR RNA upper stem-loop plays distinct roles in reverse transcription and RNA packaging. *J. Virol.* **74**, 5639-46 (2000).

19. Das, A.T., Vrolijk, M.M., Harwig, A. & Berkhout, B. Opening of the TAR hairpin in the HIV-1 genome causes aberrant RNA dimerization and packaging. *Retrovirology* **9**, 1-12 (2012).
20. Andersen, E.S. et al. Role of the trans-activation response element in dimerization of HIV-1 RNA. *J. Biol. Chem.* **279**, 22243-22249 (2004).
21. Lee, J., Dethoff, E.A. & Al-Hashimi, H.M. Invisible RNA state dynamically couples distant motifs. *Proceedings of the National Academy of Sciences* **111**, 9485-9490 (2014).
22. Jalalirad, M., Saadatmand, J. & Laughrea, M. Dominant Role of the 5' TAR Bulge in Dimerization of HIV-1 Genomic RNA, but No Evidence of TAR–TAR Kissing during in Vivo Virus Assembly. *Biochemistry* **51**, 3744-3758 (2012).
23. Moore, M.D. et al. Probing the HIV-1 genomic RNA trafficking pathway and dimerization by genetic recombination and single virion analyses. *PLoS pathogens* **5**, e1000627 (2009).
24. Chen, J. et al. HIV-1 RNA genome dimerizes on the plasma membrane in the presence of Gag protein. *Proceedings of the National Academy of Sciences* **113**, E201-E208 (2016).

REVIEWER COMMENTS

Reviewer #1 (Remarks to the Author):

An RNA excited ... at atomic resolution describes an experimental and computational strategy for elucidating rare states of RNA molecules. The approach is based on NMR relaxation experiments that are sensitive to excited conformers and that, therefore, provide structural constraints that allows ultimately to build a structure of the excited state through mutation. Once a suitable mutant is obtained that recapitulates excited state chemical shifts, computation can be used to generate ensembles that are consistent with RDC measures of the mutant. Overall the paper is very clear and the method powerful.

1) I wonder why the authors decided not to pursue a full-fledged structure determination of the mutant that mimics the excited state. Would this not be useful, even for the FARFAR-NMR approach because it would steer the choice of fragments to be those consistent with the structure. I realize that this is more work than letting a program establish potential structures based on a data-base, but given (as the authors point out) the fact that the data-base for RNA seems rather incomplete would this not be a viable approach, at least until more non-canonical base pairs populate the data-base? Or, are there so few representative fragments available in the data-base that MD is absolutely necessary? Or is it just that the NOE single structure approach would do a really poor job of recapitulating what is going on in solution where an ensemble of potentially different structures would be present?

2) A related question. How time consuming/difficult is it to use MD to generate the improved structures? And is it not important to validate what the MD is giving in terms of using more conventional NMR structural studies of the stabilized excited states. I am not suggesting doing this, but I think that the reader would appreciate why this particular strategy was chosen, when there are a number of paths that could have been taken and what the precise advantages are in the method described.

Overall, this work is a very important contribution and it is clear that a robust, easy to use protocol for obtaining these obviously important ensembles will significantly impact biology. I think that the strategy to go after the ES as an approach for regulating function is really excellent!

Reviewer #2 (Remarks to the Author):

The authors describe a procedure to determine structures of lowly populated conformations of RNA. The use site-directed mutagenesis targeted to stabilize these minor conformations of wild-type RNA sequences arguing that the minor conformation differs from the wild-type structure by alternate base pairing induce by register shift. The authors use chemical shift information from relaxation dispersion experiments to design the mutant structure, predict structures using FARFAR and cross-validate these models by experimental long-range residual dipolar coupling data.

At this stage of this investigation, I cannot recommend publication of the manuscript in Nat. Comm. for the following reasons.

- 1.) The authors use their most preferred, small model system TAR as only test case to introduce a new method. The validity of the approach to be generally applicable needs to be documented for a number of different RNA systems of different size and different levels of structural dynamics in addition to a single bulge.
- 2.) The use of FARFAR is not really cross-validated rigorously. Such cross-validation needs to be done including strange non-regular base pairs that can be exactly be the cause for stabilization of excited states. These are, however, not well predicted by FARFAR.

Reviewer #3 (Remarks to the Author):

In this manuscript the authors use NMR, computational ensembles, and MD simulations to determine structures of excited conformational states of an RNA molecule. They use HIV TAR as proof-of-principle. The technique involves trapping the excited state by mutagenesis, whose design is based on analysis of WT spectra. The idea is to then use dipolar coupling data to select for conformers derived by FARFAR-NMR. In the case of the TAR, ES2 excited state this proved challenging and further MD simulations were required to obtain a good fit and then 3D structure. Overall, this is a solid piece of work that adds to the arsenal of techniques for determining structures.

1)One obvious note that is missing is perhaps an explanation of how this technique compares to/ or is advantageous to traditional methods of using assignments, distance restraints, and RDCs to solve structures, especially given that the first step involves making a mutant that predominantly favors the excited state conformation.

3)While analysis of temperature-dependent exchange parameters on WT TAR to get the thermodynamic parameters is very sound, equivalent experiments on TAR ES2 construct are perhaps problematic given that it is stabilized by the UNCG type tetraloop. The authors need to elaborate on whether this would come into play.

4)The discussion veers towards the functional significance of ES2, which is contentious at best. It is not surprising that the mutation that favors ES2 is detrimental for transactivation, and the authors explanation for why it is not compatible for Tat and cyclin interaction makes sense. However, one would need a mutation that abolishes formation of ES2 to see if it is indeed important for any function at all

5)"Testing increasingly larger ensemble sizes (N),"
The value for N should be clearly stated in the main write up.

6)"Thus, ES2 could potentially function as a switch modulating the binding affinity of TAR to the Tat-SEC complex, possibly in a manner coupled to retroviral genome dimerization."
This is confusing and there is no reference. Does dimerization occur in the cytoplasm? How is it then coupled to transcriptional regulation in the nucleus?

7)The "outlier A-C1' chemical shifts" are hard to read in Fig 3.d

Reviewer #1 (Remarks to the Author):

I am satisfied that the authors have addressed the concerns raised by the reviewers (including performing additional experiments) and feel that the work is now acceptable for publication.

Reviewer #2 (Remarks to the Author):

The authors have addressed my previous concerns. I am happy if the paper is accepted. The authors should update the publication list with their published article:

J Am Chem Soc 2023 Oct 13 doi: [10.1021/jacs.3c04614](https://doi.org/10.1021/jacs.3c04614).

We have added the reference (number 19) mentioned by the reviewer in the main text under the introduction on page 3: “*Our NMR-based approach offers distinct advantages over X-ray crystallography and CryoEM as well as approaches employing ensemble-averaged data^{19,20} as it can determine the 3D structures of exceptionally lowly-populated (abundance <1%) and short-lived (lifetime < microsecond) ESs while also measuring their population and lifetime.*”

Reviewer #3 (Remarks to the Author):

The authors have adequately addressed my comments.